# MSDM v1.0: A machine learning model for precipitation nowcasting over East China using multisource data

Dawei Li, Yudi Liu, Chaohui Chen

Institute of Meteorology and Oceanography, National University of Defense Technology,

Changsha 410003, China; davidlilkecookies@126.com (D.L.); chenchaohui2001@nudt.edu.cn (C.C.)

*Correspondence to*: Yudi Liu (udy.liu@pku.edu.cn)

**Abstract.** East China is one of the most economically developed and densely populated areas in the world. Due to its special geographical location and climate, East China is affected by different weather systems, such as monsoons, shear lines, typhoons and extratropical cyclones. In the imminent future, the rainfall rate is difficult to predict precisely due to these systems.

Traditional physics-based methods such as numerical weather prediction (NWP) tend to perform poorly on nowcasting problems due to the spin-up issue. Moreover, various meteorological stations are distributed in this region, generating a large amount of observation data every day, which has great potential to be applied to data-driven methods. Thus, it is important to train a data-driven model from scratch that is suitable for the specific weather situation of East China. However, due to the high degrees of freedom and nonlinearity of machine learning algorithms, it is difficult to add physical constraints. Therefore,

with the intention of using various kinds of data as a proxy for physical constraints, we collected three kinds of data (radar, satellite, and precipitation data) in the flood season from 2017 to 2018 of this area and preprocessed them into tensors (256×256) that cover East China with a domain of 12.8×12.8°. The developed multisource data model (MSDM) combines the optical flow, random forest and convolutional neural network (CNN) algorithms. It treats the precipitation nowcasting task as an image-to-image problem, which takes radar and satellite data with an interval of 30 minutes as inputs and predicts radar echo

intensity with a lead time of 30 minutes. To reduce the smoothing caused by convolutions, we use the optical flow algorithm to predict satellite data in the following 120 minutes. The predicted radar echoes from the MSDM together with satellite data from the optical flow algorithm are recursively implemented in the MSDM to achieve a 120-minute lead time. The MSDM predictions are comparable to those of other baseline models with a high temporal resolution of 6 minutes. To solve blurry image problems, we applied a modified structural similarity (SSIM) index as a loss function. Furthermore, we use the random

forest algorithm with predicted radar and satellite data to estimate the rainfall rate, and the results outperform those of the traditional, nonlinear radar reflectivity factor and rainfall rate (Z-R) relationships that use logarithmic functions. The experiments confirm that machine learning with multisource data provides more reasonable predictions and reveals a better nonlinear relationship between radar echo and precipitation rate. Apart from developing complicated machine learning algorithms, exploiting the potential of multisource data will yield more improvements.

## 1. Introduction

In recent years, deep learning (DL) and machine learning (ML) have achieved great advances with big data. Tremendous meteorological data are produced every day, which perfectly matches these novel data-driven artificial intelligence (AI) approaches. Quantitative precipitation nowcasting (QPN) using radar echo extrapolation (REE) has recently become popular (Tran and Song, 2019). Precipitation nowcasting predicts rainfall intensity in the following few hours. Based on various data with high spatiotemporal resolutions, AI precipitation prediction can be relatively accurate compared with traditional numerical weather prediction (NWP) methods. U-Net (Ronneberger et al., 2015) is a well-known network designed for image segmentation, and its core is upsampling, downsampling, and skip connection. It can efficiently achieve high accuracy with a small number of samples. Agrawal et al.(2019) treated precipitation nowcasting as an image-to-image problem. They employed U-Net (Ronneberger et al., 2015) to predict the change in radar echo for QPN, which is superior to High Resolution Rapid Refresh (HRRR) numerical prediction from the National Oceanic and Atmospheric Administration (NOAA) when the prediction time is within 6 hours. Sonderby et al.(2020) proposed a neural weather model (NWM) called MetNet that uses axis self-attention (Ho et al., 2019) to discover weather patterns from radar and satellite data. MetNet can predict the next 8 hours of precipitation in 2-minute intervals with a resolution of 1 kilometer. Shi et al. (2015) treated precipitation nowcasting as a problem of predicting spatiotemporal sequences and modified the fully connected long short-term memory (FC-LSTM) by replacing the Hadamard product with a convolution operation in the input-to-state and state-to-state transitions. They believe that cloud movement is highly uniform in some areas, and convolutions can capture these local characteristics. Therefore, the convolution operation in the input transformations and recurrent transformations of their proposed convolutional LSTM (ConvLSTM) helps to handle the spatial correlations. Furthermore, they apply the same modification to the gated recurrent unit (GRU) and notice that convolution is location-invariant and focuses on only a fixed location because its hyperparameters (kernel size, padding, dilation) are fixed. However, in the QPN problem, a specific location of cloud clusters continuously changes over time. Hence, Shi et al. (2017) proposed a trajectory GRU (TrajGRU) that uses a subnetwork to output a location-variant connection structure before state transitions. The dynamically changed connections help TrajGRU capture the trajectory of cloud clusters more accurately than previous methods. In the field of video prediction, Wang et al. proposed various recurrent neural networks (RNNs) based on LSTM. For example, they designed PredRNN++ (Wang et al., 2018) with a cascaded dual memory structure and gradient highway unit, which strengthens the power for modeling short-term dynamics and alleviates the vanishing gradient problem, respectively. In addition, to capture spatial characteristics through recurrent state transitions, Wang et al. (2019a) integrated 3D convolutions inside LSTM units and proposed Eidetic 3D LSTM (E3D-LSTM). Moreover, Wang et al. (2019b) designed the memory in memory (MIM) network to handle higher-order nonstationarity of spatiotemporal data. By using differential signals, MIM can model the nonstationary properties between adjacent recurrent states. However, their work is based on a slight modification of existing techniques demanding massive computing resources for model training and has not been applied to big meteorological data.

Computer vision techniques have long been used in object detection, video prediction, and human motion prediction. Tran and Song (2019) used image quality assessment techniques as a new loss function instead of the common mean squared error (MSE), which misled the process of training and generated blurry images. Ayzel et al. (2019) designed an advanced model based on the multiple optical flow algorithm for QPN, but it still performs poorly in the prediction of the onset and decay of precipitation systems because optical flow methods simply calculate the position and velocity of the radar echo with a constant velocity rather than consider the changing intensity of radar echo.

On the one hand, the current massive amounts of data are underutilized; on the other hand, scientists in the field of machine learning focus on pursuing high accuracy by increasing the complexity of models based on a single source of data. Given this background, from the perspective of atmospheric science, we build a multisource data model (MSDM) with the aim of fully using multisource observation data (for example, radar reflectivity, infrared satellite data, and rain gauge data) and find suitable machine learning algorithms (for example, deep neural network, optical flow, and random forest algorithms) for each type of data that can ensure accuracy while saving computing resources. In addition, due to the high degrees of freedom and nonlinearity of neural networks, it is difficult to apply physical constraints to these machine learning models. Hence, we hope that multisource data will function as a proxy for physical constraints to guide the model during the training process. The main advantage of MSDM lies in its transferability: any machine learning model and observation data can be incorporated into the model. For example, wind speed data can be a proxy for dynamic constraints, and temperature data can function as a proxy for thermodynamic constraints. Due to the limit of computing resources, the aim of this paper is not to achieve a higher resolution or prediction accuracy but to propose a method combining machine learning and deep learning with radar echo data, satellite data, and automatic ground observation data to achieve physically reasonable QPN.

Section 2 introduce the related work about the use of machine learning and deep learning models for radar and precipitation. The dataset, models and methods used in this study are described in section 3. Section 4 shows the results. Section 5 draws conclusions and discusses some possible future work.

## 2. Related work

### 2.1 Machine learning

There is a large volume of published studies describing the use of ML for radar and precipitation. Logistic regression, as one of the simple ML algorithms, has been used to improve the forecast of precipitation probability (Vislocky and Young, 1989). Kuligowski and Barros (1998) use neural networks to postprocess NWP output and forecast precipitation in the next 6 hours. Lakshmanan et al. (2014) use neural networks to improve quality control of weather radar data. K-means clustering is a form of unsupervised learning, which is used for the classification of precipitation (Yang and Deng, 2010). Hwang et al. (2019) modify this kind of clustering algorithm and train two nonlinear regression models to improve the subseasonal forecast of temperature and precipitation. Support vector machine (SVM) uses kernels to transform data to the nonlinear space, which has been applied to forecast tornadoes (Adrianto et al., 2009), predict precipitation in tropical cyclones(Wei, 2012), and train

precipitation estimation model (Huang et al., 2015). Decision tree algorithms are widely used in classification and regression tasks. Gagne et al. (2009) use the decision tree to classify storm types based on radar observations. Loken et al (2019) calibrate the ensemble precipitation forecast via random forest. Hill et al. (2020) use random forests to predict the probability of severe weather across the United States. Mao and Sorteberg (2020) use random forest to train a binary model to improve the accuracy of radar-based precipitation nowcasts. Bayesian technique is an important branch of ML. Todini (2001) use it to improve the radar precipitation estimation. Fox and Wikle (2005) propose a quantitative precipitation nowcast scheme based on Bayesian hierarchical model. Chandra et al. (2021) use Bayesian machine learning to reconstruct annual precipitation from climate-sensitive lithologies and improve the predictive accuracy of global circulation models (GCM) at a low computational cost.

## 2.2 Deep learning

Deep learning (DL; LeCun et al., 2015) has gained popularity in meteorology recently. The existing literature on the application of DL to radar and precipitation is extensive. Foresti et al. (2019) train artificial neural networks (ANN) on a 10-year archive of radar images in Swiss to nowcast the growth and decay of precipitation. Sadeghi et al. (2019) apply CNN together with the same kind of data to estimate precipitation, which shows great improvement compared with baseline models. Yan et al. (2021) introduce a Flow-Deformation network (FDNet) that captures the motion of the optical flow field and the deformation of radar echoes at the same time. The ML community tend to treat nowcasting problems as the prediction of spatiotemporal sequences. Chen et al. (2020) use ConvLSTM for nowcasting and early warning of heavy rainfall. Ran et al. (2021) use Faster-RCNN (Ren et al., 2016) to identify precipitation clouds for doppler weather radar. The deep neural networks have also been applied to reduce the bias and false alarms of satellite-based precipitation products (Tao et al., 2016). Tao et al. (2017) design two DL models that incorporate satellite data from infrared and water vapor channels to identify the precipitation, which significantly outperform the operational product. Yo et al. (2020) propose a volume-to-point framework for radar-based QPE, which can automatically detect the movement and evolution of precipitation systems. Ravuri et al. (2021) present a deep generative model to eliminate the blurry nowcast at longer lead times. As for data fusion, Chandra and Kapoor (2020) design a Bayesian transfer learning framework to provide an approach for handling multiple sources of data. Veillette et al. (2020) use satellite data, radar image, and lightning flash data to synthetic weather radar. Miao et al. (2020) deem the nowcasting problem as a computer vision task and propose a multimodal graph framework to model different data jointly.

## 3. Materials and Methods

### 3.1 Dataset

The spatial and temporal distribution characteristics of precipitation are related to many factors, such as the terrain, atmospheric circulation, and climatic conditions. To train a deep learning model that can capture the precipitation characteristics of East China, we collected multisource observation data of the flood season (May to September) for a total of 306 days from 2017 to

2018. Due to the missing radar data from May 1 to 9 and September 26 to 30, 2018, there are only 292 days of radar data in total. The missing data are obtained by interpolating the data from adjacent moments. Among the data, the precipitation data of regional automatic weather stations (AWS) in East China with a time interval of 10 minutes are shown in Fig 1.

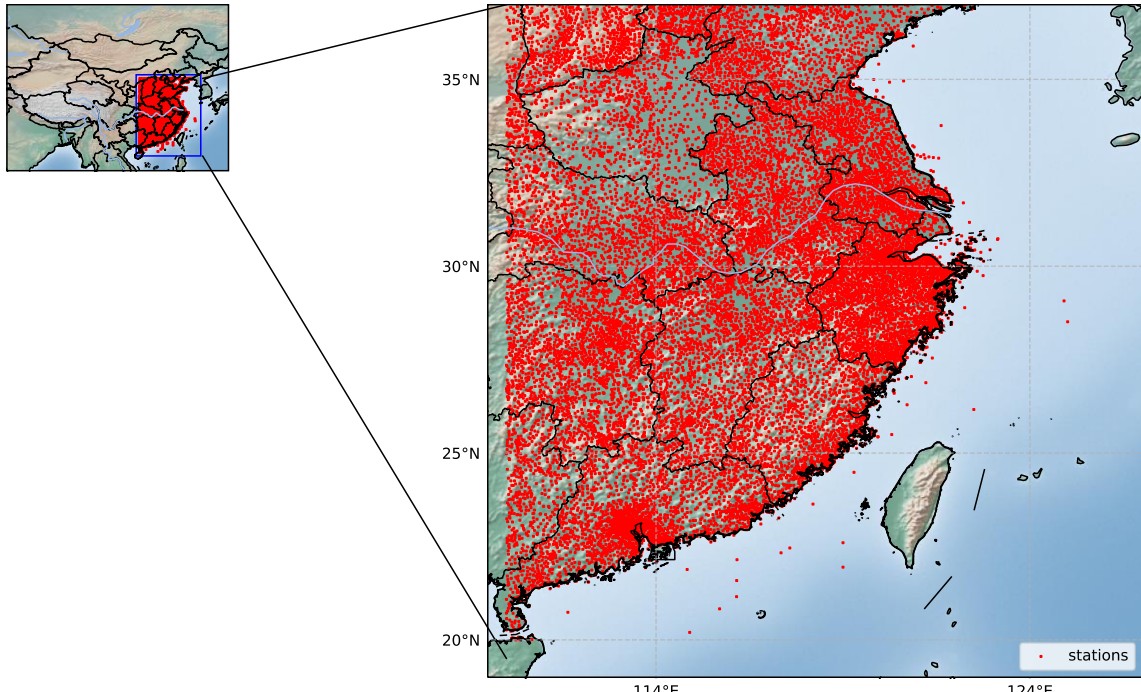

**Figure 1. Distribution of the automatic ground stations in East China.**

The weather radar data (resolution of 0.5°×0.5°) have been preprocessed into the combined reflectivity; the latitude range is from 21.0°N to 36.0°N, the longitude range is from 112.0°E to 125.9°E, and data were available every 6 minutes (Fig 2(a)). The Himawari 8 satellite brightness temperature data (resolution 0.5° × 0.5°) for channels 07-16 are used with a latitude range of 19-37°N, a longitude range of 110-127°E, and a time interval of 30 minutes (Fig 2(b)). The links for the datasets are as follows:

Radar data: http://data.cma.cn/data/detail/dataCode/J.0012.0003.html,

AWS data: http://data.cma.cn/data/detail/dataCode/A.0012.0001.html,

Himawari 8 satellite data: http://www.cr.chiba-u.jp/databases/GEO/H8_9/FD/index.html.

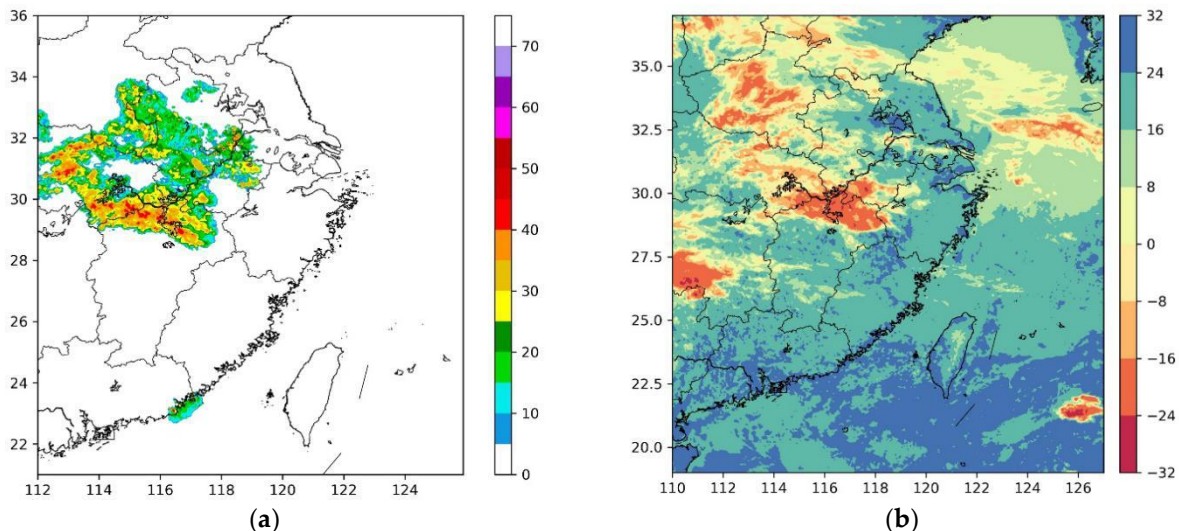

**Figure 2. Combined reflectivity (unit: dBZ) in East China (a), Himawari 8 satellite brightness temperature data (unit: °C) of channel 07 (b) on May 1, 2017.**


## 3.2 Model Description

### 3.2.1 Model Architecture

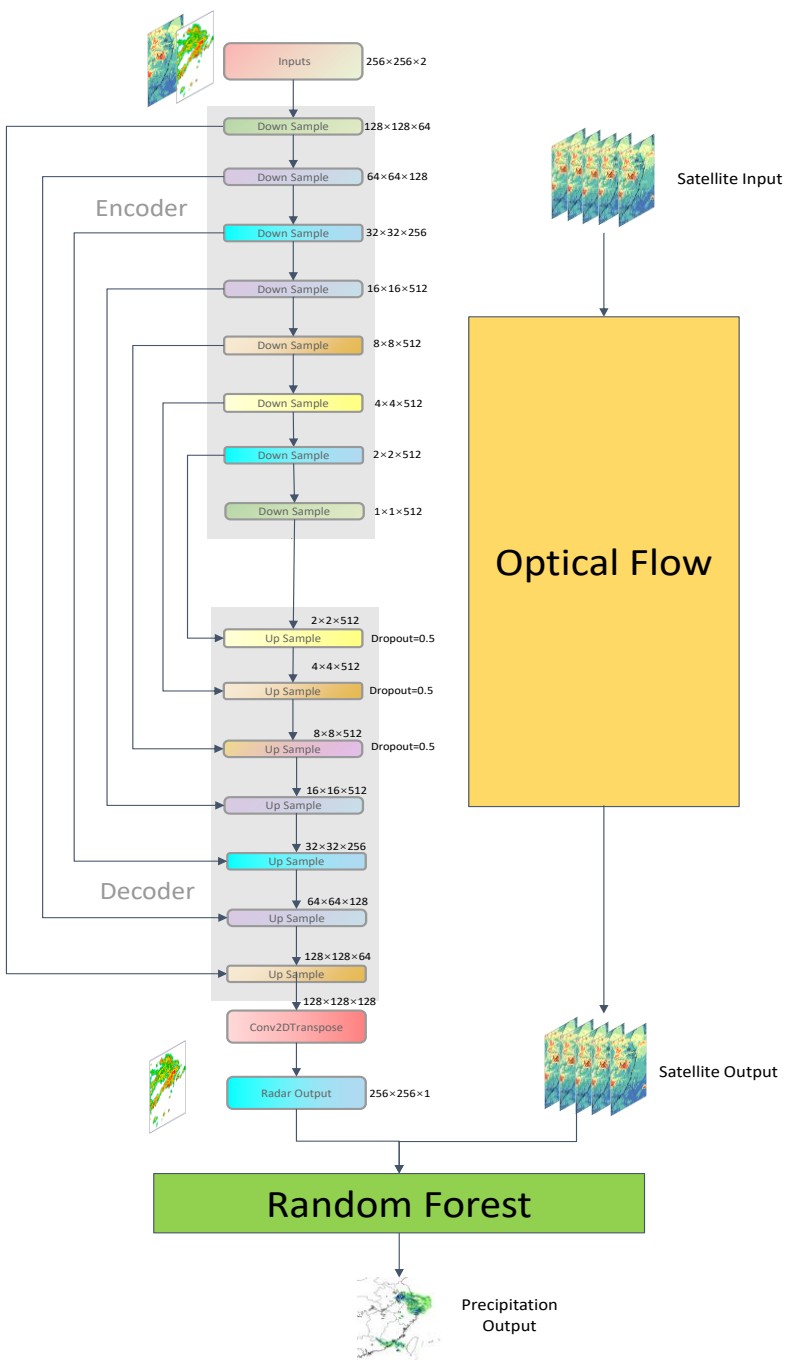

**Figure 3. Structure of the MSDM.**

To incorporate multisource data, we designed an MSDM with three parts: deep learning, optical flow, and random forest (Fig 3). The deep learning part of the MSDM is inspired by the state-of-the-art U-Net (Ronneberger et al., 2015) designed for image segmentation. It follows the encoder-decoder structure that encoder has 8 down sample blocks and decoder has 7 up sample blocks. Each downsampling block in the encoder consists of Conv2D, batch normalization, and leaky rectified linear unit (LeakyReLU) activation layers. Each upsampling block in the decoder includes transposed convolutional, batch normalization,

dropout of 0.5 (applied to the first 3 blocks), and ReLU activation layers. In each convolutional layer, the step size parameter (stride) is set to 2, and padding is set to 'same'. The kernel size varies between 4×4 and 2×2 to extract the spatial characteristics at different scales. The batch normalization layer effectively avoids the gradient disappearance problem and improves the convergence speed. We use dropout to randomly discard some information with a probability of 50% to prevent overfitting. The activation function adds nonlinearity to each block and allows the model to better learn the nonlinear relationship between

the input and target. Transposed convolutional layers are introduced to substitute upsampling layers in U-Net to increase the resolution of the images. As in U-Net, there are skip connections between the encoder and decoder to solve the problem of gradient explosion and gradient disappearance during training.

    The primary reason that we use transposed convolutional layers to replace upsampling layers is that both layers are used for upsampling images. Upsampling layers use an interpolation method (for example, nearest neighbor interpolation, bilinear

interpolation, and bicubic interpolation) to rescale the input image to a desired size with a higher resolution. These interpolation methods are preset, so there is little room for the network to learn. The deconvolution operation is not a predefined interpolation method, and it has some learnable parameters to convert the output to the original image resolution. Through the training of the model, it will learn an optimal upsampling method instead of a preset method.

    In the deep learning part, the MSDM takes the array with a shape of 256×256×2, which represents the height, width and

channel of the image. Radar and satellite grid point data are at different channels. The output of this part is a predicted radar image 30 minutes later with a shape of 256×256×1. The optical flow part takes 5 consecutive satellite frames as input to extrapolate the satellite image in the following 2 hours. Subsequently, the predicted radar image and satellite image will be used in two parts. First, it will flow into the random forest part to estimate the precipitation rate. Second, it will be recursively used as the input of the deep learning part to achieve a lead time of two hours.

The reasons why we do not predict precipitation directly using deep learning are as follows: 1) The precipitation data we collected are irregular site data, which are distributed only on land and do not include precipitation on the sea (Fig 1). The combined radar reflectivity (Fig 2(a)) and Himawari 8 satellite data (Fig 2(b)) are regular grid point data and include sea data. The spatial distributions of these three types of data are inconsistent, so it is impossible to make a feature-label correspondence to directly predict precipitation. 2) The use of shapefiles to extract radar echo or satellite data on land will cause the edge of

the echo to be limited to the land, which loses the meaning of extrapolation. 3) We hope to improve the transferability of MSDM that can integrate different kinds of data except grid point data. Therefore, the method of processing precipitation data can be used on other observation site data in daily operation. 4) We believe that deep learning efficiently extracts the long-

period trend in precipitation, but it cannot capture the transient characteristics of precipitation. Therefore, for each rainfall event, we use random forest to model the nonlinear relationship between multisource data to capture its unique characteristics.

### 3.2.2 Reference Models

#### 3.2.2.1 Optical flow method

We first employed rainy motion v1, an optical flow model proposed by Ayzel et al (2019), to evaluate the performance of the optical flow algorithm for tracking and extrapolating radar echoes by our dataset. It performs poorly on the radar echo data when the lead time is up to 60 minutes. However, it performs better on satellite data, which are recorded every 30 minutes. We believe that cloud layer motion is dominated by air advection transportation; thus, the optical flow method can better simulate its motion characteristics. Additionally, the temporal resolution of satellite data is coarser (30 minutes), so we can directly obtain the sequence of four frames of the following 2 hours through one prediction rather than iterative prediction. Optical flow can predict such short sequences quickly and shows great advantages in saving computing resources and avoiding error accumulation. In addition, the main drawback of the convolution operation is that it smooths the characteristics of the image, and the level of smoothness increases when applying convolutions recursively in deep learning models. Therefore, to ease the smoothing of radar echoes and preserve more details of precipitation systems, we decide to use the results of satellite data predicted by the optical flow component of our model.

#### 3.2.2.2 ConvLSTM

ConvLSTM (Shi et al., 2015) is a traditional model for the QPN problem. Hence, we compare our model with ConvLSTM to see whether the model with multisource data performs well when we simply formulate QPN as an image-to-image problem rather than a spatiotemporal sequence problem (Eq. 1).

$$\widetilde{\mathcal{X}}_{t+1}, \dots, \widetilde{\mathcal{X}}_{t+5} = \underset{\mathcal{X}_{t+1}, \dots, \mathcal{X}_{t+5}}{argmax} \, p(\mathcal{X}_{t+1}, \dots, \mathcal{X}_{t+5} \mid \mathcal{X}_{t-5+1}, \mathcal{X}_{t-5+2}, \dots, \mathcal{X}_t), \quad (1)$$

Tensor $\mathcal{X}_t$ represents the radar echo map in the shape of 256×256 at time t, and tensor $\widetilde{\mathcal{X}}_{t+1}$ represents the model prediction result.

#### 3.2.2.3 U-Net

U-Net (Ronneberger et al., 2015) was employed by Agrawal et al. (2019) for QPN. They treat the problem as an image-to-image problem (Eq. 2) to forecast the precipitation in the next hour.

$$\widetilde{\mathcal{X}}_{t+5} = \underset{\mathcal{X}_{t+5}}{argmax} \, p(\mathcal{X}_{t+5} \mid \mathcal{X}_t), \quad (2)$$

Tensor $\mathcal{X}_t$ and $\widetilde{\mathcal{X}}_t$ is as in Eq. 1, we use the U-Net architecture to predict the radar image 30 minutes later in comparison to the MSDM to demonstrate that the combination of multisource data is better than single source data.

### 3.3 Training and evaluation method of the MSDM

The model that we designed is a modified U-Net model (Fig 3). We use the radar and satellite data as inputs, and the output is the intensity of the radar echo after half an hour (Eq. 3). The two kinds of data were fed into the encoder and then concatenated by skip connections and flowed into the decoder and transposed convolutional layer (Fig 3).

$$\widetilde{\mathcal{X}}_{t+5} = \underset{\mathcal{X}_{t+5}}{argmax}\, p(\mathcal{X}_{t+5} \mid \mathcal{X}_t, \mathcal{Y}_t), \tag{3}$$

The MSDM uses weather radar echo data $\mathcal{X}_t$ and Himawari 8 satellite brightness temperature data $\mathcal{Y}_t$ to predict the radar echo map at time t+5. After the first round of prediction, we combined $\widetilde{\mathcal{X}}_{t+5}$ from our model and the predictions of $\widetilde{\mathcal{Y}}_{t+5}$ from optical flow for further prediction. During preprocessing, the weather radar data and Himawari 8 satellite brightness temperature data are extracted, which cover the area of 23.0-35.8°N, 113.0-125.8°E with a 256×256 window. Then, the values

of these data $Z$ are transformed into pixels $P$ by Eq. 4

$$P = \frac{Z - min\{Z\}}{max\{Z\} - min\{Z\}}, \tag{4}$$

To improve the image quality, we apply a modified structural similarity index (SSIM) (Wang et al., 2004) as the loss function, which is helpful to solve blurry image problems. The loss function for the predicted image and ground truth is defined by Eq. 5:

$$Loss = -1 \times SSIM(y_{pred}, y_{true}) = -1 \times \frac{\left(2\mu_{y_{pred}}\mu_{y_{true}} + C_1\right)\left(2\sigma_{y_{pred}y_{true}} + C_2\right)}{\left(\mu_{y_{pred}}^2 + \mu_{y_{true}}^2 + C_1\right)\left(\sigma_{y_{pred}}^2 + \sigma_{y_{true}}^2 + C_2\right)}, \tag{5}$$

where $y_{pred}$ is the predicted image, $y_{true}$ is the ground truth, and $\mu_{y_{pred}}$ and $\mu_{y_{true}}$ are the average values of $y_{pred}$ and $y_{true}$, respectively. $\sigma_{y_{pred}}^2$ and $\sigma_{y_{true}}^2$ are the variances of $y_{pred}$ and $y_{true}$, respectively. $\sigma_{y_{pred}y_{true}}$ is the cross-correlation of $y_{pred}$ and $y_{true}$. $C_1$ and $C_2$ are small positive constants. In each calculation, a window of 3×3 is taken from the image, and then the

window is continuously sliding for calculation. Finally, the average value is taken as the global SSIM.

To evaluate our model, a comparison was made between the optical flow method, ConvLSTM, and U-Net methods. Due to limits on computational resources, we use a few frames to predict the results for the half-hour. Then, the output results are used to iteratively predict the radar echo in the next half-hour to achieve a lead time of 2 hours (Fig 4). For the baseline sequence-to-sequence models (ConvLSTM, optical flow), we use the first 5 frames ($T_{-4}$~$T_0$) to predict a sequence of the next

5 frames ($T_1$~$T_5$) and use this result to iteratively predict the remaining three sequences ($T_6$~$T_{10}$, $T_{11}$~$T_{15}$, $T_{16}$~$T_{20}$). For image-to-image models (U-Net, MSDM), we use frame $T_0$ to predict frame $T_5$ and use this prediction as input to iteratively predict the following frames ($T_{10}$, $T_{15}$, $T_{20}$).

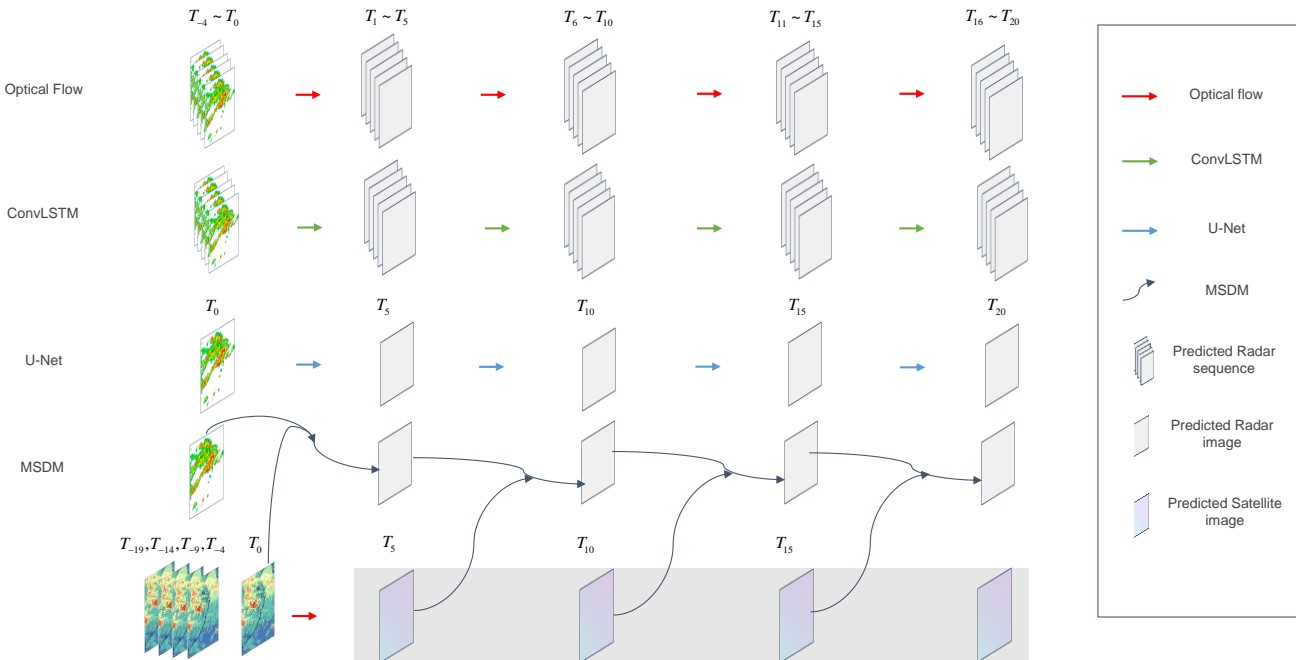

**Figure 4. The time sequences of the optical flow, ConvLSTM, U-Net and MSDM.**

## 3.4 Performance Evaluation

The MSDM is trained with our dataset on Google Colab Pro with TensorFlow-GPU-2.2.0 and executed on an NVIDIA Tesla P100 GPU (16 GB). In total, 240 days of data used for training, 26 days for validation and 26 days for testing. All the models are compiled with the Adam optimizer, and the learning rate is set at 0.001. To avoid overfitting, we apply the early-stopping strategy to monitor the loss in the validation set. We use several metrics to evaluate the model's performance on the test set, namely, the critical success index (CSI, Eq. 6), Heide skill score (HSS, Eq. 7), false alarm ratio (FAR, Eq. 8) (Woo and Wong, 2017), root mean square errors (RMSEs), and use the SSIM to evaluate the structural similarity between the generated image and target image.

$$CSI = \frac{hits}{hits + misses + false\ alarms}, \qquad (6)$$

$$HSS = \frac{2(\,hit \cdot correct\ negative - miss \cdot false\ alarm\,)}{miss^2 + false\ alarm^2 + 2 \cdot hit \cdot correct\ negative + (\,miss + false\ alarm\,)(\,hit + correct\ negative\,)}, \qquad (7)$$

$$FAR = \frac{false\ alarm}{hit + false\ alarm}, \qquad (8)$$

where the correct negatives, hits, misses and false alarms are determined by the threshold value. Woo and Wong (2017) provide more details about these metrics. We applied six thresholds of 0.1, 1, 5, 10, 25, and 40 dBZ to calculate the CSI, HSS and FAR. To stress the importance of areas with large radar reflectivity, we assign a weight w( threshold ) (Eq. 9) to different thresholds and calculate the weighted CSI and HSS (the larger the better).

$$w(\text{threshold}) = \begin{cases} 1, & \text{threshold} = 0.1 \\ 1, & \text{threshold} = 1 \\ 2, & \text{threshold} = 5 \\ 3, & \text{threshold} = 10 \\ 5, & \text{threshold} = 25 \\ 8, & \text{threshold} = 40 \end{cases} \quad\quad\quad\quad (9)$$

We set all the weights to 1 for the FAR (the smaller the better) because we believe that the influence of false alarms of every threshold is the same. The RMSE is used to evaluate the global error of the predicted radar image. For the SSIM, we set the Gaussian filter size to 3×3 and the width to 1.5 to evaluate the local structural similarity between the generated image and target image.

## 4. Results

### 4.1 REE

In the region we select over East China, the radar echo and precipitating cloud system change little between two adjacent frames (6 minutes). Therefore, the results of all the models are shown every 30 minutes (Fig 5). The input of optical flow and ConvLSTM is a sequence of 5 frames before time 0, and the output is a sequence of 5 frames in the following half-hour. The
input of U-Net is a single frame of the radar echo data at time 0, and the input of the MSDM includes a frame of satellite data and a frame of the radar echo data. When the output of the first 30 minutes is obtained, we take it as the input to replace the real data for further prediction. After the first step of prediction, the satellite data are input into the MSDM for QPN by the optical flow algorithm. Because cloud movements are dominated by advective motion, the optical flow method is used.

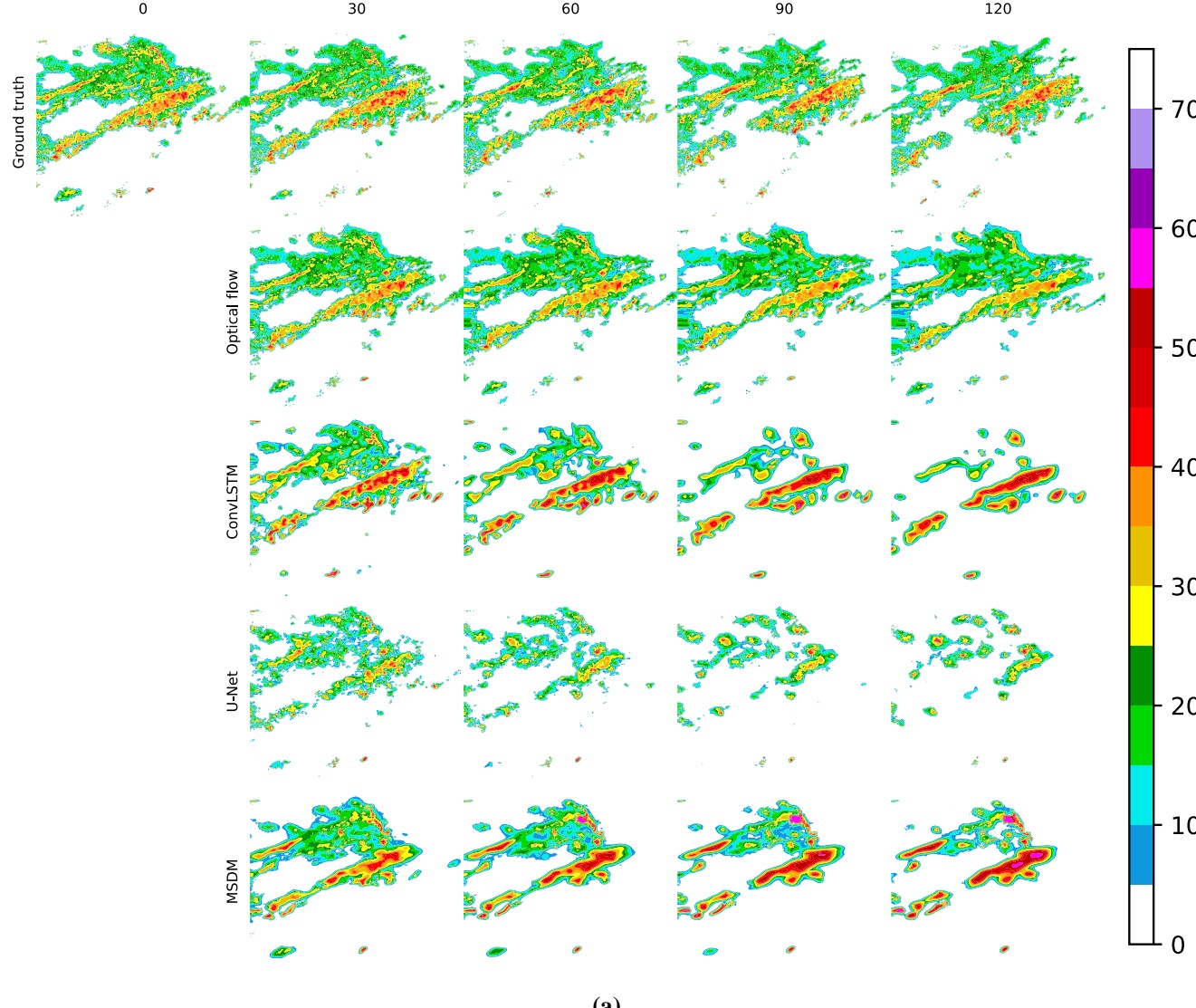


**(a)**

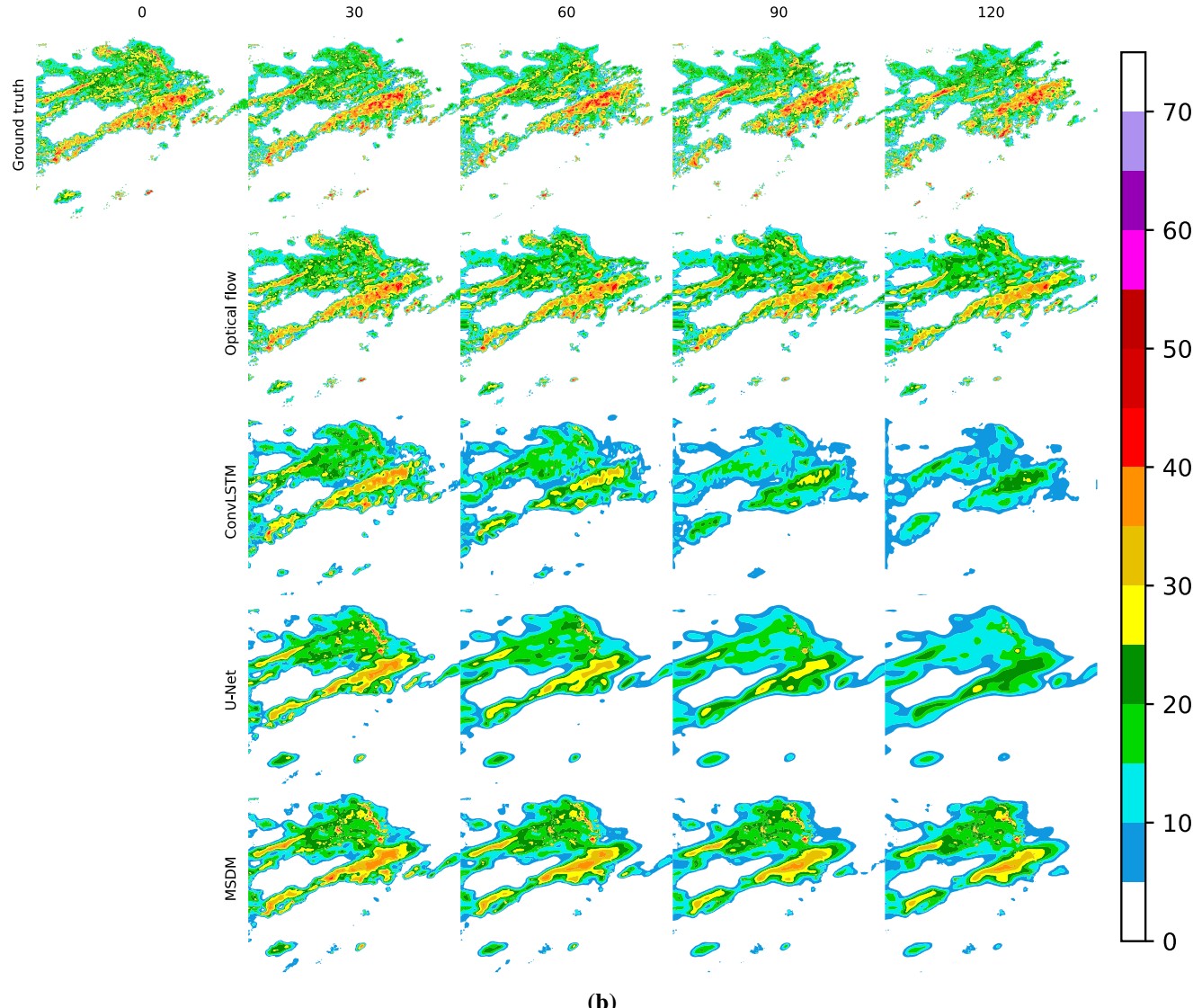

**(b)**

**Figure 5. Illustrations of the observed radar echo, the radar echo simulated by the optical flow, ConvLSTM, U-Net and MSDM. For the optical flow and ConvLSTM, we select one frame every half-hour for comparison with other models. In Fig 5(a), each model was trained with the modified SSIM. In Fig 5(b) each model was trained with the MSE. The date and time are September 7, 2018, 00:00.**

In Fig 5, we present the comparison of 4 models trained with different loss functions. Fig 5(a) shows that the models trained with the modified SSIM predict many large-value areas of radar echo because the SSIM can extract the local structural similarity through the training process. In contrast, Fig 5(b) shows that models trained with the MSE tend to smooth the details of radar echo and seldom predict large radar echo values because the large-value area is only a small part of the entire echo, and the MSE will ignore these areas when it optimizes errors on a global scale. Hence, the modified SSIM shows its advantage when compared with the conventional loss function in the REE task.

The radar echoes predicted by the ConvLSTM, U-Net and MSDM decay in the following 2 hours, while those predicted by the optical flow method remain stable. Thus, the optical flow method can perfectly predict the edge and shape of the radar echo, which is the reason why it obtains the highest average weighted CSI at a lead time of 30 minutes (Table 1) on the testing set. However, the fatal weakness of the optical flow method is that it simply predicts radar echo movement from previous images without predicting radar echo decay and initiation, which causes its accuracy to decrease over time (Table 1), and the FAR keeps increasing (Table 3). In addition, it employs an algorithm called a corner detector (Ayzel et al., 2019) to identify special points from previous frames and track the movement of these points. When it extrapolates the tail of the radar echo, it cannot find corresponding points from previous images because the tail of the radar echo at this moment was in a position outside the radar image of previous frames. Consequently, unreasonable shapes exist in the tail of the predicted radar echo. In Fig 5(a), we find that ConvLSTM performs the best for the strong echoes, but it cannot maintain the shape of the echo. Additionally, there exists a phenomenon in which only the strong-echo area is increasing, while the weak-echo area is continuously decreasing, which is contradictory according to fluid continuity theory. The ConvLSTM captures the temporal features from previous frames, which strengthens the intensity, but it cannot properly predict the initiation and decay of the whole system. This finding could explain why it obtains the lowest FAR in the last hour (Table 3) because the fewer the number of predicted echoes, the lower the ratio of making mistakes is.

ConvLSTM is prone to error accumulation due to iterative training and requires massive computing resources (Yu et al., 2018). Therefore, we use a convolutional neural network (CNN) as a substitute to treat REE as an image-to-image problem. U-Net along with our MSDM can generally simulate the motion of the radar echo while maintaining its outline, but the MSDM with satellite data can avoid radar echo decay through iterations. The MSDM has comparable performance with baseline models and outperforms other models in the short-term period (Table 1 and Table 2). We believe it retains the merits of the optical flow method, which can maintain the shape of the radar echo, and it has the ability to predict the strong-echo area from U-Net. The MSDM performs poorly when the lead time is longer than 90 minutes because the cumulative error from the two kinds of data was larger than either of both. In addition, satellite data may provide more details that the radar echo may not contain, for example, data over the sea; instead, these details may be treated as noise or false alarms, so the accuracy will decrease.

Table 1 and Table 2 show the weighted average CSI and HSS on the test set with different thresholds (0.1, 1, 5, 10, 25, 40, unit: dBZ). The two metrics are used to evaluate the performance of each model (the higher the better). From Table 1, we notice that optical flow method achieves the best score when the lead time is 30 minutes, which shows its great advantage in short-term forecasting. However, its long-term predictions are not accurate due to a lack of simulation of the radar echo evolution. ConvLSTM performs poorly because it increases only the strong echo but neglects the prediction of low-value areas. Hence, even though it obtains high scores on large reflectivity areas, its weighted CSI and HSS are still lower than those of the other models. U-Net also performs poorly due to its inability to handle temporal correlations and the absence of key spatial information. The MSDMs with different loss functions (MSE and SSIM) perform well in long-term forecasting. The SSIM can capture the structural similarities of radar images, while the MSE can calculate the global errors. However, SSIM is prone to error accumulation through iterative prediction. Therefore, in Table 1, MSDM_ssim ranks best for lead times of 60 minutes and 90 minutes, while MSDM_mse ranks best for other lead times. Satellite data add more spatial information for the MSDM to learn and set physical constraints on it. Therefore, the MSDM best scores in the first three moments of the weighted HSS. Regarding the

FAR, the MSDM still performs best in the first two moments due to its reasonable prediction of the shape and intensity of the radar echoes. ConvLSTM ranks best in the last two moments because it forecasts only strong echoes of a few areas, which greatly reduces the probability of false alarms.


**Table 1.** Weighted average CSI on the test set with different thresholds (0.1, 1, 5, 10, 25, 40, unit: dBZ). The best scores are highlighted in bold. The second-best score is underscored (the larger the better).

| Model | 30 min | 60 min | 90 min | 120 min |
|---|---|---|---|---|
| Optical Flow | **0.414** | 0.303 | 0.209 | 0.205 |
| ConvLSTM | 0.399 | 0.269 | 0.211 | 0.157 |
| U-Net | 0.348 | 0.259 | 0.216 | 0.184 |
| MSDM_mse | 0.362 | 0.286 | 0.245 | **0.218** |
| MSDM_ssim | 0.405 | **0.317** | **0.258** | 0.217 |

**Table 2.** Weighted average HSS on the test set with different thresholds (0.1, 1, 5, 10, 25, 40, unit: dBZ). The best scores are highlighted in bold. The second-best score is underscored (the larger the better).

| Model | 30 min | 60 min | 90 min | 120 min |
|---|---|---|---|---|
| Optical Flow | 0.512 | 0.409 | 0.34 | **0.304** |
| ConvLSTM | 0.487 | 0.311 | 0.246 | 0.18 |
| U-Net | 0423 | 0.307 | 0.25 | 0.209 |
| MSDM_mse | 0.437 | 0.341 | 0.29 | 0.255 |
| MSDM_ssim | **0.514** | **0.413** | **0.343** | 0.291 |


**Table 3.** Average FAR on the test set with different thresholds (0.1, 1, 5, 10, 25, 40, unit: dBZ). The best scores are highlighted in bold. The second-best score is underscored (the smaller the better).

| Model | 30 min | 60 min | 90 min | 120 min |
|---|---|---|---|---|
| Optical Flow | 0.316 | 0.391 | 0.439 | 0.474 |
| ConvLSTM | 0.265 | 0.295 | **0.242** | **0.246** |
| U-Net | 0.293 | 0.309 | 0.313 | 0.309 |
| MSDM_mse | 0.329 | 0.364 | 0.387 | 0.399 |
| MSDM_ssim | **0.237** | **0.27** | 0.303 | 0.335 |

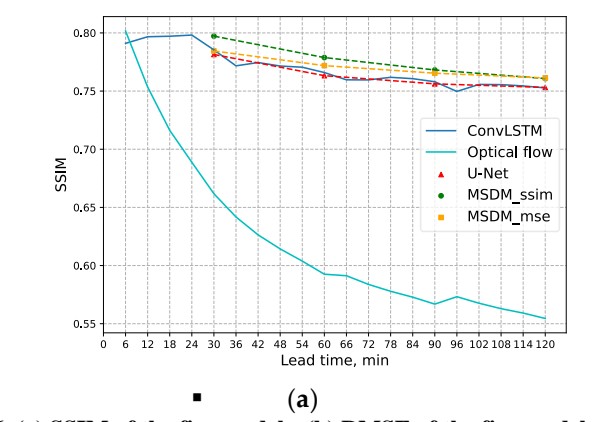 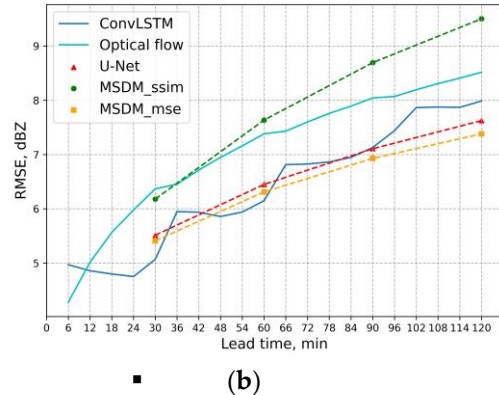

(a)  (b)

**Figure 6. (a) SSIM of the five models. (b) RMSE of the five models, dBZ.**

We calculate the SSIM and RMSE between the predicted radar echoes of the four models and the ground truth on the test set (Fig 6). The optical flow model achieves the lowest SSIM (Fig 6a), which means that it has the worst SSIM to the ground truth. MSDM_ssim obtains the highest score on the SSIM but the worst performance on the RMSE because it focuses on only local features but ignores the minimization of the global error. ConvLSTM, U-Net, and MSDM_mse are trained on the MSE loss function, which achieve a lower RMSE. We believe that when the SSIM is used as the loss function, the model will

generate more reasonable predictions with proper shapes, but it will lead to poor performance on global evaluation metrics such as the mean absolute error (MAE) and RMSE. Moreover, we notice that the ConvLSTM model produces larger errors in the first frame of each sequence than other models. This phenomenon can result from the deficiency of LSTM that cannot handle accumulative error, which is magnified by iterative prediction.

## 4.2 QPN

Previous works seem to pay little attention to QPN after they achieve good performance on REE tasks. Researchers tend to use an empirical formula to calculate the precipitation rate based on the prediction of radar echo from models. Shi et al. (2015) employed the Z-R relationship ($Z = 10 \log a + 10b \log R$) to calculate the rainfall, where Z represents the radar echo in the dBZ, R represents the rainfall rate in mm/h, and a and b are two constants that are calculated based on the statistical data of specific regions. We believe that this empirical formulation cannot describe the nonlinear relationship between the radar echo

intensity and the rainfall rate. Therefore, random forest machine learning regression techniques are used to describe this relationship. The weather radar data and precipitation data one hour before the prediction time are used for training. The method we take is as follows. First, an automatic station is identified. Then, the radar and satellite data for these grid points as well as the corresponding rainfall rate from site points are applied to train the random forest model. Finally, the learned nonlinear relationship is used to predict the rainfall rate an hour later.

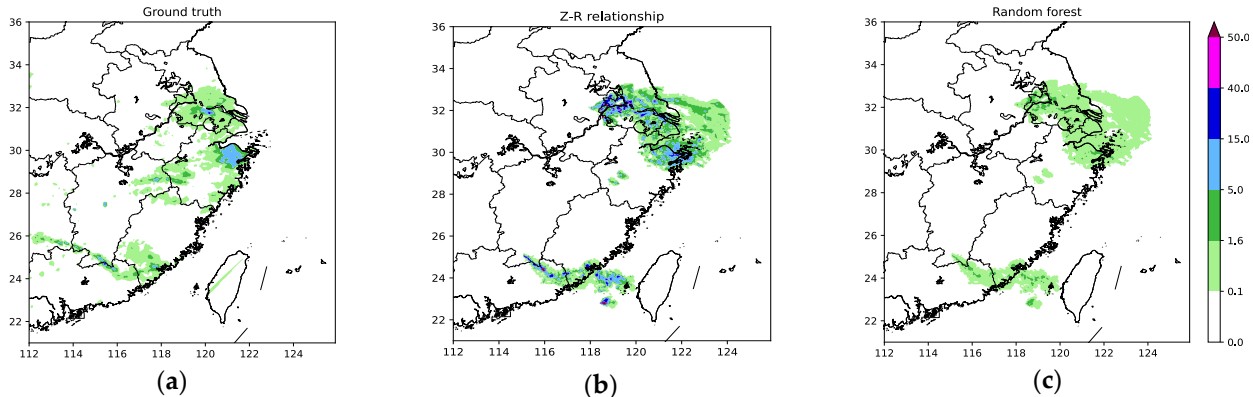

Figure 7. (a) Ground truth interpolated from site points, mm/h. (b) Rainfall rate calculated by the Z-R relationship, mm/h. (c) Rainfall rate calculated by the random forest model, mm/h

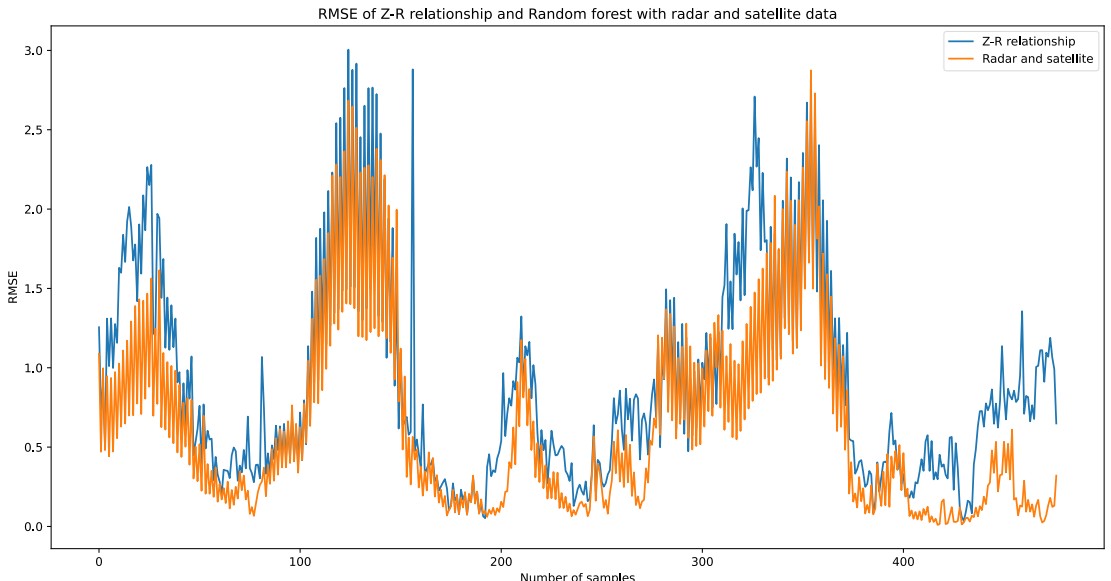

Figure 8. RMSE of 480 QPN samples predicted by the two methods

Figure 7 shows the results of the Z-R relationship and random forest model. Since the precipitation data on the grid points are obtained by interpolation and might have errors, we did not make a quantitative comparison for the whole dataset. However, this example shows that the Z-R relationship tends to overestimate the rain intensity. For example, the Z-R relationship predicts many areas with precipitation rates larger than 15 mm/h, but there are few areas that reach the value on the ground truth. Figure 8 shows the RMSEs of 480 QPN samples using different methods and data. When we use the radar and satellite data as input, the random forest model shows its superiority in the QPN task. Its RMSEs are lower than those of the Z-R relationship in most of the samples. Therefore, we believe that multisource data have great potential to make the results more precise.

# 5. Conclusions and discussion

 ## 5.1 Discussion

In Table 4, we evaluate four models in terms of 12 performance indictors (amount of data required for training, time needed for training the model, false alarm rate, cumulative system error, ability to capture spatial/temporal characteristics, ability to predict the radar echo initiation and decay, 0~1 hour forecast accuracy, 1~2 hour forecast accuracy, ability to maintain the radar echo shape, clarity of the radar image, conforming to the laws of physics). We use the mark '↓' to represent that the

 lower the better and the mark '↑' to represent the higher the better. Subsequently, we discuss and summarize the advantages and limitations of the models and their combinations.

**Table 4.** Evaluation on four models with the performance indictors

| | Amount of data required for training ↓ | Time needed for training the model ↓ | False alarm rate ↓ | Cumulative system error ↓ |
|---|---|---|---|---|
| Optical flow | 1 | 1 | 2 | 1 |
| ConvLSTM | 4 | 4 | 3 | 2 |
| U-Net | 2 | 2 | 4 | 2 |
| MSDM | 3 | 3 | 1 | 4 |

| | Ability to capture spatial characteristics ↑ | Ability to capture temporal characteristics ↑ | Ability to predict radar echo initiation and decay ↑ | 0~1 hour forecast accuracy ↑ |
|---|---|---|---|---|
| Optical flow | 1 | 3 | 1 | 3 |
| ConvLSTM | 2 | 4 | 2 | 1 |
| U-Net | 3 | 1 | 3 | 2 |
| MSDM | 4 | 1 | 4 | 4 |

| | 1~2 hour forecast accuracy ↑ | Ability to maintain the radar echo shape ↑ | Clarity of the radar image ↑ | Conforming to the laws of physics ↑ |
|---|---|---|---|---|
| Optical flow | 1 | 4 | 4 | 4 |
| ConvLSTM | 4 | 1 | 1 | 1 |

| | | | | |
|---|---|---|---|---|
| U-Net | 3 | 2 | 2 | 2 |
| MSDM | 2 | 3 | 3 | 3 |

Here, the smaller the first four indictors values are (amount of data required for training, time needed for training the model, false alarm rate, cumulative system error), the better the model performance is; the larger the last eight indictors values are (ability to capture spatial/temporal characteristics, ability to predict the radar echo initiation and decay, 0~1 hour forecast accuracy, 1~2 hour forecast accuracy, ability to maintain the radar echo shape, clarity of the radar image, conforming to the laws of physics), the better the model performance is. Form Table 4 we can see that all of them have advantages and

disadvantages. We are going to discuss the strong points and weak points of the methods.

Optical Flow:

The advantages of the optical flow algorithm are as follows: (1) It has the fewest parameters and takes the least time to train. (2) The amount of data required for training the model is small, and at least 2 radar images can be used to extrapolate the radar echo. (3) It maintains the shape of the radar echo very well, and the prediction result is closest to the real echo. Therefore, its

MSE is the smallest. (4) It is suitable for the extrapolation of advection precipitation from 0 to 1 hour in the future.

The disadvantages of the optical flow algorithm are as follows: (1) It cannot extract features of the evolution process of the radar echo. (2) Except the advective precipitation, it performs poorly in other precipitation situations (e.g. convective precipitation and typhoon precipitation), in which the radar reflectivity changes rapidly in a short period of time. For the large-value area of radar echo, it basically has no forecasting ability. (3) The tail of the echo cannot be extrapolated due to the lack

of previous data. As a result, the longer the lead time, the more irregular the shape of the echo at the tail.

ConvLSTM:

The advantages of ConvLSTM are as follows: (1) It can extract the spatial characteristics of echoes while capturing the time characteristics efficiently. (2) It can simulate the initiation and decay of radar echo better than optical flow. (3). It is the best for the prediction of long time and large-value areas of radar echo.

The disadvantages of ConvLSTM are as follows: (1) There are many parameters, many matrix operation and various gating structures in ConvLSTM. Therefore, its training speed is the slowest among the four models. (2) It overestimated/underestimated the large/lowvalue radar echo, which does not conform to the fluid continuity theory. (3) It predicts the worst shape of the echo in that there is no transition between the large echo area and the nonecho area, which is far away from the true echo and has no guidance for operational forecasting. For example, we cannot issue an early warning

of heavy precipitation in one place, and at the same time it cannot forecast if there will be no rain in its neighboring places.

U-Net:

The advantages of U-Net are as follows: (1) It is an efficient CNN that has relatively few parameters and can achieve high accuracy with a small amount of data. (2) It is capable in capturing the spatial characteristics of radar echoes and predicting the evolution of echoes. (3) The forecasting effect is very good for the next one or two frames.

The disadvantages of U-Net are as follows: (1) It is unable to extract the temporal characteristics of changes in the radar echo. (2) The convolution operation will smooth the characteristics of the radar echo so that the shape of the predicted echo will change and deviate from the true one. (3) Through iterative training and prediction, the error accumulates.

## 5.2 Conclusions

As a conventional QPN method, the optical flow method has played a certain role in the forecasting of advective precipitation. However, it performs poorly in the prediction of advective precipitation due to the simplicity of its algorithm and the lack of use of existing big data (Woo and Wong, 2017). Moreover, deep learning shows great advantages in processing vast amounts of data. By using convolution and LSTM structures, deep learning algorithms are better at capturing spatiotemporal correlations. Nevertheless, recurrent networks (represented by ConvLSTM) for predicting spatiotemporal sequences are

widely known to be difficult to train and computationally expensive (Yu et al., 2018). Compared with traditional spatiotemporal sequence tasks in the field of machine learning, such as moving Modified National Institute of Standards and Technology (MNIST) prediction, human position prediction, and traffic flow prediction, the REE task has specific background and physical constraints. Therefore, merely obtaining predictions with higher scores does not reflect the quality of the results. Wang et al. (2018,2019) designed state-of-the-art models to capture comprehensive correlations between spatiotemporal

sequences. However, when we apply them to the physics-based tasks represented by REE and QPN, we must evaluate their prediction from the perspective of atmospheric science. The prediction is of reference significance only when it is physically reasonable rather than having high scores. However, it is difficult to apply physical constraints to neural networks due to their high degrees of freedom and nonlinearity. Hence, we input more kinds of data as features into the network with the intention that it can obtain more information through feature interaction. Therefore, we collect multisource data and design an MSDM.

In a situation in which when the model becomes incorrect and tries to predict low radar reflectivity, the incorporated satellite data will balance it. We hope the multisource data function as another form of model constraint. Solving the sequence-to-sequence problem is computationally expensive, so we treat the QPN as an image-to-image problem and design the MSDM based on a CNN (U-Net) with high efficiency and few parameters. The main advantage of the MSDM is its transferability. Apart from satellite data, any other data (wind speed, pressure, temperature, etc.) can be used as input into the model in the

future. Wind speed data could add dynamic constraints, and temperature data could add thermodynamic constraints. To further save computational resources, we use optical flow to predict the sequence of satellite data with the assumption that the cloud cluster is dominated by convective movement. This approach is adopted by an operational nowcasting system to estimate convective cloud movement (Shi et al., 2017). Subsequently, we use the satellite data predicted by optical flow and radar reflectivity predicted by the MSDM as input for iterative prediction to achieve a lead time of 2 hours. After predicting the

radar echo, we replace the empirical formula (Z-R relationships) with a random forest model to estimate the rainfall rate. We believe that deep learning models capture the long-term trend in precipitation. There should be an algorithm that captures real-

time dynamic characteristics, and random forest regression is very suitable for short-term prediction with small samples. Therefore, we trained a random forest regressor using radar and precipitation data from one hour prior. Subsequently, the learned nonlinear relationships were applied to estimate the precipitation rate from radar reflectivity.

In conclusion, the MSDM combines the merits of optical flow and U-Net, maintains the pattern of the radar echo, and predicts their initiation and decay. The results predicted by the MSDM also contain more details that U-Net cannot produce. Given the background that ConvLSTM overestimates the strong echo and underestimates the weak echo, the MSDM shows great potential in predicting areas of both strong and weak radar echo. We conducted an experiment by using random forest for QPN, which obtained relatively better results than those obtained by the Z-R relationship. This finding suggests that the

empirical formula is not suitable for all areas. We believe that by the combination of multisource data, the radar echoes predicted by the MSDM can provide more details and have more physical constraints than those predicted by single-observation data. It not only learns the long-term trend through deep learning but also incorporates real-time dynamic characteristics captured by the optical flow and random forest models. Hence, the prediction from the MSDM is more physically reasonable and of reference significance.

Currently, methods still exist to estimate the precipitation rate more precisely. For example, Wu et al (2020) used a graph convolutional regression network to produce more spatial characteristics of precipitation. For future works, we believe that the predictions could be more accurate with RNNs and GRUs. Additionally, the precipitation rate should consider the influence of the terrain and different scales. In fact, we will perform further experiments on these factors.

*Code and data availability.* The source code and pretrained model of MSDM are available at http://doi.org/10.5281/zenodo.4749183.

*Author contribution.* Conceptualization, D.L., Y.L. and C.C.; methodology, software, investigation, D.L.; resources, data curation, C.C.; writing—original draft preparation, D.L.; writing—review and editing, Y.L.; visualization, C.C.; supervision, Y.L.; project administration, C.C.; funding acquisition, Y.L. All authors have read and agreed to the published version of the

manuscript.

*Competing interests.* The authors declare that they have no conflict of interest.

*Acknowledgements.* This research is supported by the National Natural Science Foundation of China (41875060)

*Review statement.* This paper was edited by Rohitash Chandra and reviewed by Patrick Armand and three anonymous referees.

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
