# Peer review of "MSDM v1.0: A machine learning model for precipitation nowcasting over East China using multisource data"

_Geoscientific Model Development, 2020_

## Short Comment (SC1) · 26 Feb 2021

Dear authors,

in my role as Executive editor of GMD, I would like to bring to your attention our Editorial version 1.2:

https://www.geosci-model-dev.net/12/2215/2019/

This highlights some requirements of papers published in GMD, which is also available on the GMD website in the 'Manuscript Types' section: http://www.geoscientific-model-development.net/submission/manuscript_types.html

In particular, please note that for your paper, the following requirement has not been met in the Discussions paper:

- "The main paper must give the model name and version number (or other unique identifier) in the title."

- Code must be published on a persistent public archive with a unique identifier for the exact model version described in the paper or uploaded to the supplement, unless this is impossible for reasons beyond the control of authors. All papers must include a section, at the end of the paper, entitled "Code availability". Here, either instructions for obtaining the code, or the reasons why the code is not available should be clearly stated. It is preferred for the code to be uploaded as a supplement or to be made available at a data repository with an associated DOI (digital object identifier) for the exact model version described in the paper. Alternatively, for established models, there may be an existing means of accessing the code through a particular system. In this case, there must exist a means of permanently accessing the precise model version described in the paper. In some cases, authors may prefer to put models on their own website, or to act as a point of contact for obtaining the code. Given the impermanence of websites and email addresses, this is not encouraged, and authors should consider improving the availability with a more permanent arrangement. Making code available through personal websites or via email contact to the authors is not sufficient. After the paper is accepted the model archive should be updated to include a link to the GMD paper.

Please provide the version number of MSDM in the title of your revised manuscript.

As GitHub is not a persistent archive, please provide a persistent release for the exact source code version of rainymotion v1 used for the publication in this paper. As explained in https://www.geoscientific-model-development.net/about/manuscript_types.html the preferred reference to this release

is through the use of a DOI which then can be cited in the paper. For projects in GitHub a DOI for a released code version can easily be created using Zenodo, see https://guides.github.com/activities/citable-code/ for details.

Finally note, that according to our new Editorial (v1.2) all data and analysis / plotting scripts should be made available.

Even more important, note that google drive is not acceptable to be used for data / code provision for a publications in GMD. You have to provide the data in a more persitent archive.

Yours, Astrid Kerkweg

———————————————

---

## Referee Comment (RC1) · Anonymous Referee #1 · 4 Mar 2021

Precipitation nowcasting is important as it provides valuable insights into drought and flood risk management. This has been achieved by the authors using multi-source data model. The authors seem to make a novel contribution with the study, however the paper warrants further improvement. Recommendation is to do a thorough copy editing of the paper, and include a discussion section to examine the relevance of your results and how it relates to other studies. See below for specific comments.

Abstract

L9 – Change to 'predict precisely'

L13 – insert 'is' between that and suitable
L13 – change collect to collected

L22 – Define SSIM

L23 – explain the traditional Z-R relationship

Introduction

L33 – Remove extra citation

L34 – What is NOAA's HRRR? There is a need to go through the entire paper and define all the acronyms in its first use, even for something as common as AI.

L35 – Remove repeated citation

L35 – What is MetNet ? Do not assume the reader knows

L37 – Remove repeated citation and do this for every other instance where this is the case

L37 – How were Shi.et al able to achieve this ? Explain their method of prediction of spatio-temporal predictions

L41 – Avoid the use of contractions eg. Haven't

L43 – Avoid using etc. if you can not name more items

L44 – L45 – incorrect use of tense, past (used) and present (will mislead). Check for grammatical errors throughout the paper

L45 – How is optical flow method related to Trans method ? Or is it not related ? Why is it mentioned.

L51 – Remove et.al

Materials and Methods

L59 – Rephrase "to train the deep learning model to learn"

L66 – Provide links for datasets/sources

L74 – change "we compared" to a comparison was made

L75 – What exactly is being predicted first? Needs more clarity

L84 – change to "makes it better to predict". The entire paper needs to be thoroughly edited for English writing and grammar.

L84 – Are you trying to say that the satellite data is more coarse ? Or has low spatial resolution ? Or are you referring to the time resolution ?

L87-88 – Explain more what you mean by " increases level through recursive application" and justify your use of satellite data .

Results

L120-124 – There is no mention of how the model was trained and the results were validated. Ideally this information should be in the methods

L120 – It almost appears that the aim of the paper has not been clearly stated. You are using multiple sources of data and at the same time creating a multi-source data model (MSDM). This is very difficult to follow throughout the paper. Make this distinction clear

L125 – Use other metrices to evaluate model performance

L148 – Who made that claim? Citation ?

Conclusions and Discussions

L195-210 - Include a thorough discussion to examine the relevance of your results and how it relates to other studies, previous methods used etc..

L195 – Can it be conclusively stated that looking at the problem through image-image prediction is better than focusing on the problem as spatio-temporal sequence problem. If that is the aim of your paper, then what is the conclusion ? Deliberate further.

Quick Report

1. Does the paper address relevant scientific modelling questions within the scope of GMD? Does the paper present a model, advances in modelling science, or a modelling protocol that is suitable for addressing relevant scientific questions within the scope of EGU? - Yes

2. Does the paper present novel concepts, ideas, tools, or data? - Yes

3. Does the paper represent a sufficiently substantial advance in modelling science? Yes

4. Are the methods and assumptions valid and clearly outlined? The method is valid but needs to be clearly outlined and requires more justifications why certain approaches were taken.

5. Are the results sufficient to support the interpretations and conclusions? - Yes

6. Is the description sufficiently complete and precise to allow their reproduction by fellow scientists (traceability of results)? In the case of model description papers, it should in theory be possible for an independent scientist to construct a model that, while not necessarily numerically identical, will produce scientifically equivalent results. Model development papers should be similarly reproducible. For MIP and benchmarking papers, it should be possible for the protocol to be precisely reproduced for an independent model. Descriptions of numerical advances should be precisely reproducible. – The method needs to be explained more clearly.

7. Do the authors give proper credit to related work and clearly indicate their own new/original contribution? – Yes, however the authors need to discuss their work in relation to previous work to illustrate how this study contributes to the bigger picture

8. Does the title clearly reflect the contents of the paper? The model name and number should be included in papers that deal with only one model.- Yes

9. Does the abstract provide a concise and complete summary?- Yes

10. Is the overall presentation well structured and clear? - Yes

11. Is the language fluent and precise? The writing is clunky in some places and needs editing, corrections.

12. Are mathematical formulae, symbols, abbreviations, and units correctly defined and used? - Yes

13. Should any parts of the paper (text, formulae, figures, tables) be clarified, reduced, combined, or eliminated?- No

14. Are the number and quality of references appropriate? – Could benefit from more references

15. Is the amount and quality of supplementary material appropriate? For model description papers, authors are strongly encouraged to submit supplementary material containing the model code and a user manual. For development, technical, and benchmarking papers, the submission of code to perform calculations described in the text is strongly encouraged.- Yes

---

## Referee Comment (RC2) · Anonymous Referee #2 · 4 Mar 2021

Overview: In this paper, Li et al. developed a machine learning model (MSDM) for precipitation nowcasting using multi-source data, and compared the new model with existing methods. The critical success indexes (CSI) calculated from the results of MSDM are comparable with other methods. However, the root mean squared error (RMSE) of MSDM is greater than Optical flow and ConvLSTM, indicating that the error of MSDM is greater than other models. In this case, it can be concluded that the overall performance of this new model (MSDM) is not as good as some existing models. The model presented in this paper is new, but its performance has not been demonstrated to be better than existing models. In this case, I suggest the authors to further modify and train the model, and submit a modified version after its overall performance (judged

by RMSE) reaches existing models.

Specific comments: (1) The critical success indexes (CSI) depends on the artificial threshold chosen by the authors, making it difficult to judge whether MSDM performs better than Optical flow and ConvLSTM. The authors might choose several thresholds (i. e., 0.1, 0.3, 1, 3, 10, 40), and calculate the average CSI of these thresholds, then we can compare these models more easily. (2) RMSE is used frequently to judge the performance of machine learning models, but the RMSE of MSDM is too high compared with optical flow, so I suggest the authors to improve and re-train the MSDM model to get a lower RMSE. (3) The MSDM should be described in more details. In Fig. 3, the red arrow on "ours" (MSDM) indicates that the optical flow is used in MSDM, but in Fig. 4 there is no optical flow in the structure of MSDM? (4) In this paper, the authors used MSDM to predict radar echo, and then used random forest to predict precipitation. Why not predict precipitation directly using MSDM? Theoretically, the error would be smaller if one single network is used to predict precipitation.

---

## Short Comment (SC2) · 4 Mar 2021

1.Comments from referees: Please provide the version number of MSDM in the title of your revised manuscript. Author's response: We will replace 'MSDM' with 'MSDM v1.0' in the title in our revised manuscript.

2. Comments from referees: You have to provide the data in a more persistent archive. Author's response: We will upload our data and analysis / plotting scripts to Zenodo and provide a DOI in the revised manuscript. The source code of rainy motion v1 will also be transferred to Zenodo. The data that we downloaded from China Meteorological Data Service Centre (http://data.cma.cn/en) are not allowed to transfer or for external

use without the permission of the meteorological authority. We will provide links for researchers to apply and obtain these data.

---

## Referee Comment (RC3) · P. Armand (Referee) · 6 Mar 2021

The paper by Li, Liu, and Chen is entitled "MSDM: a machine learning model for precipitation nowcasting over East-China using multi-source data".

As mentioned by the authors, the imminent rainfall rate may be difficult to predict and Numerical Weather Prediction (NWP) sometimes perform poorly for the nowcasting (notably due to the spin-up issue). Otherwise, numerous meteorological stations are available and can be used with data-driven methods.

In this work, three kinds of data (radar, satellite, precipitation) have been collected during the 2017 and 2018 flood season in a domain of 12.8° x 12.8° covering East-China. The authors have developed a "Multi-Source Data Model (MSDM)" that combines AI methods (optical flow, Convolutional Neural Network, and random forest).

The MSDM considers the precipitation nowcasting task as an image-to-image problem. The authors take radar and satellite data with a interval of 30 minutes as inputs in order to predict radar echo intensity with a lead time of 30 minutes. To reduce the smoothing caused by the convolution, they use optical flow to predict satellite data in the following 120 minutes. The predicted radar echo from MSDM together with satellite data from optical flow are recursively implemented to achieve 120 minutes lead time. Moreover, the authors use random forest with predicted radar and satellite data to estimate the rainfall rate.

The authors show that the MSDM predictions are comparable to those of the baseline models with a high temporal resolution of 6 minutes. They argue that machine learning with multi-source data provides more reasonable predictions and reveals a better non-linear relationship between radar echo and rainfall rate in comparison with a sole source of data. Still, improvements in the algorithms developed by the authors seem necessary.

OVERALL COMMENTS The paper is correctly structured and written. It is interesting as it relates to data-driven methods of AI leveraged for numerical weather nowcasting. I agree that a better rainfall rate forecast would be of high interest for risk prevention (here, flooding prevention). The principal flaw of the paper is that the methods and their parameters are insufficiently described. Thus, it is extremely difficult to evaluate the scientific accuracy of the paper and the quality of the results. Furthermore, it is very surprising that "Multi-Source Data Model (MSDM)" has together the preference of the authors and, by far, not the best scores compared to other methods. There is certainly an explanation, but it does not straightforwardly appear in the paper. This explanation should be given in the paper. Thus, the paper could be published after major revisions.

SPECIFIC COMMENTS L 034 - What does the HRRR acronym stand for?

L 035 - Explain what are the "U-Net" and "Met-Net" methods.

L 038 - Explain what is a TrajGRU model.

L 040 - Some concise comments about the PredRNN++, MIM, and E3D-LSTM networks are necessary.

L 065 - The Figure 1 is not legible. You should improve it.

L 122 - I wonder if 240 days of data are enough to train the MSDM. Is this choice explained by a limitation in the computations or is there another justification?

L 135 - Figure 5 corresponds to a particular date and time. The authors should indicate what they are on the figure. Moreover, I wonder what would be the results for other dates and times. There are too few results presented for the validation and test of the AI methods. More results should be shown.

L 142 - I do not understand: "it tracks features by the corner detector". What does it mean?

L 156 - Table 1 - I guess that the Critical Sucess Index is given for four methods, but only for one date and time. What about other test-cases? I think that the methods should be benchmarked in a large number of situations in order to be able to comment the scores.

Otherwise, the MSDM ranks very differently depending on the observation times (from 30 to 120 minutes) with the 0.1 dBZ threshold. Is it logical and explainable? Is it worth noticing that the MSDM ranks consistently (second best score) with the 40 dBZ thresholds. What would be the scores of the MSDM for the Radar Echo Extrapolation at other dates and times?

L 165 - It seems that using the Modified Structural Similarity Index (denoted by SSIM) is counter-productive in terms of MAE and RMSE. Why use it? Once more, I wonder if

the example of results produced for one date and time has a general value.

L 186 - Figure 8 - Looking at this figure, I am not very convinced that the CSI of the Quantitative Precipitation Nowcasting are better using the random forest than using the Z-R relationship. In general, the scores are quite similar. Could the authors try to better advocate the random forest method?

L 212 - The acronyms "RNN" and "GRU" should be developed.

My general feeling about the AI methods used separately or combined together through the paper is that all of them have advantages and drawbacks. I suggest to the authors to add a final synthetic table describing the strong points and weak points of the methods and of their combinations. This would greatly help the readers to understand the arguments of the authors.

---

## Referee Comment (RC4) · Anonymous Referee #4 · 7 Mar 2021

This manuscript discusses the potential of some ML methods to tackle the nowcasting problem. It is an interesting paper and it can become a valuable contribution. However, it requires modifications and improvements before being acceptable for publication.

First of all, it is unclear why the authors prefer the multi-model method when it does not give the best results in comparison with others. Furthermore, to me this method just seems like a combination of all other possible methods: Optical flow, Random forest and Convolutional Neural Network (CNN). Can one really disentangle what contributions to the solution of the problem are coming from each of the components? More insight into this is needed, especially if the method is not the one leading to the best

results.

Some minor comments are:

Parentheses and periods misplaced. Incorrect double citations throughout the paper Many acronyms that are never defined. 11. its -> their or the 12-13. check sentence 14. ndarray? 28. Unnecessary 'the' 74-75. Due to limits on computational resources Figure 6 increase the labels It is not very scientific to label the method being tested as 'ours'
* * *

---

## Author Comment (AC1) · 23 Apr 2021

Thank you for your interest in our paper and for your helpful remarks. We have corrected structural and grammar issues. The model architecture, training and evaluating method, and discussion about relevance of our results have been added in the revised manuscript. The point-by-point responses to all comments are as follows:

1. L9 – Change to 'predict precisely'
Response: The change has been made.
Changes in the manuscript: 'in the imminent future the rainfall rate affected by which is difficult to precisely predict precisely'

2. L13 – insert 'is' between that and suitable
Response: The change has been made.
Changes in the manuscript: 'it is important to train a data-driven model from scratch that is suitable to…'

3. L13 – change collect to collected
Response: The change has been made.
Changes in the manuscript: 'We collected three kinds of data (radar, satellite, precipitation) in flood season…'

4. L22 – Define SSIM
Response: We develop the acronyms SSIM as Structural Similarity (SSIM) and the definition of which is in eq.4:

$$Loss = -1 \times SSIM(y_{pred}, y_{true}) = -1 \times \frac{\left(2\mu_{y_{pred}}\mu_{y_{true}}+C_1\right)\left(2\sigma_{y_{pred}y_{true}}+C_2\right)}{\left(\mu_{y_{pred}}^2+\mu_{y_{true}}^2+C_1\right)\left(\sigma_{y_{pred}}^2+\sigma_{y_{true}}^2+C_2\right)}$$

Changes in the manuscript: 'we applied a modified Structural Similarity (SSIM) index as a loss function.'

5. L23 – explain the traditional Z-R relationship
Response: The change has been made.
Changes in the manuscript: 'the results outperform those of the traditional Z-R relationships that use logarithmic function to describe the non-linear relationships between radar reflectivity and rainfall rate'

6. What is NOAA's HRRR? There is a need to go through the entire paper and define all the acronyms in its first use, even for something as common as AI.
Response: National Oceanic and Atmospheric Administration (NOAA) High Resolution Rapid Refresh (HRRR). Other acronyms have been defined.
Changes in the manuscript: 'which is superior to High Resolution Rapid Refresh (HRRR) numerical prediction from National Oceanic and Atmospheric Administration (NOAA) when the prediction time is within 6 hours.'

7. L35 – Remove repeated citation

Response: The change has been made.
Changes in the manuscript: 'Sonderby et al.(2020) proposed'

8. L35 – What is MetNet? Do not assume the reader knows
Response: We have explained the 'MetNet' in the revised manuscript.
Changes in the manuscript: 'Sonderby et al.(2020) proposed a Neural Weather Model(NWM) called MetNet that uses axis self-attention (Ho et al., 2019) to discover the weather pattern from radar and satellite data. MetNet can predict the next 8 hours precipitation with a resolution of 1 kilometer in 2-minute intervals.'

9. L37 – Remove repeated citation and do this for every other instance where this is the case
Response: The change has been made and the entire paper has been thoroughly edited for English writing and grammar.
Changes in the manuscript: 'Shi et al.(2015)'

10. L37 – How were Shi.et al able to achieve this? Explain their method of prediction of spatio-temporal predictions
Response: We have explained their method in the revised manuscript.
Changes in the manuscript: 'Shi.et al (2015) treated the precipitation nowcasting as a problem of predicting spatio-temporal sequence and modified the fully-connected Long Short-Term Memory (FC-LSTM) by replacing the hadamard product with convolution operation in the input-to-state and state-to-state transitions. They believe that cloud movement is highly uniform in some areas, and convolution can capture these local characteristics. Therefore, the convolution operation in the input transformations and recurrent transformations of their proposed Convolutional Long Short-Term Memory (ConvLSTM) helps to handle the spatial correlations.'

11. L41 – Avoid the use of contractions eg. Haven't
Response: The change has been made.
Changes in the manuscript: 'and have not been applied to the numerous meteorological data'

12. L43 – Avoid using etc. if you cannot name more items
Response: The change has been made.
Changes in the manuscript: 'Computer vision techniques have long been used in object detection, video prediction, and human motion prediction.'

13. L44 – L45 – incorrect use of tense, past (used) and present (will mislead). Check for grammatical errors throughout the paper
Response: The use of tense has been corrected and the entire paper has been proof read by an English first language checker.
Changes in the manuscript: 'Song(2019) used image quality assessment techniques as a new loss function instead of the common mean squared error(MSE), which misled

the process of training and generate the blurry image.'

14. L45 – How is optical flow method related to Trans method? Or is it not related? Why is it mentioned.

Response: It is related to Ayzel et al 's work and has been moved to the right part.

Changes in the manuscript: 'Ayzel et al. (2019) designed an advanced model based on the multiple optical flow algorithm for QPN, but it still performs badly in the prediction of onset and decay of precipitation systems because Optical flow methods simply calculate the position and velocity of the radar echo with a constant velocity rather than consider the changing intensity of radar echo.'

15. L51 – Remove et.al

Response: We have removed 'et.al'

Changes in the manuscript: 'Given this background, from the perspective of atmospheric science, we build a multi-source data model (MSDM) with the aim to fully use multi-source observation data (for example, radar reflectivity, infrared satellite data, and rain gauge data)'

16. L59 – Rephrase "to train the deep learning model to learn"

Response: It has been rephrased.

Changes in the manuscript: 'To train a Deep learning model that can capture the precipitation characteristics of East China'.

17. L66 – Provide links for datasets/sources

Response: The links have been provided in the revised manuscript.

Changes in the manuscript:

'Radar data: http://data.cma.cn/data/detail/dataCode/J.0012.0003.html , AWS data: http://data.cma.cn/data/detail/dataCode/A.0012.0001.html , Himawari 8 satellite data: http://www.cr.chiba-u.jp/databases/GEO/H8_9/FD/index.html '

18. L74 – change "we compared" to a comparison was made

Response: The change has been made.

Changes in the manuscript: 'To test our method, comparison was made…'

19. L75 – What exactly is being predicted first? Needs more clarity

Response: We have clarified in the revised manuscript.

Changes in the manuscript: 'Due to limits on computational resource, we use few frames to predict the results in half an hour. Then, the output results are used to iteratively predict the radar echo in the next half an hour to achieve a lead time of 2 hours (Fig 4). For the baseline sequence-to-sequence models (ConvLSTM, Optical flow), we use first 5 frames ($T_{-4} \sim T_0$) to predict a sequence of the next 5 frames($T_1 \sim T_5$), and use this result to iteratively predict the remaining three sequences ($T_6 \sim T_{10}$, $T_{11} \sim T_{15}$, $T_{16} \sim T_{20}$). For image-to-image models (U-Net, MSDM), we use frame $T_0$ to predict frame $T_5$, and use this result as input to iteratively predict the following frames ($T_{10}$,

$T_{15}$, $T_{20}$).'

20. L84 – change to "makes it better to predict". The entire paper needs to be thoroughly edited for English writing and grammar.
Response: The change has been made and the entire paper has been thoroughly edited for English writing and grammar.
Changes in manuscript: 'make it better to predict…'

21. L84 – Are you trying to say that the satellite data is more coarse ? Or has low spatial resolution? Or are you referring to the time resolution?
Response: Yes, the temporal resolution of satellite data is more coarse. Its interval is 30 minutes. We try to express that it only has four frames in the lead time of 2 hours. Therefore, we can get the sequence of the following 2 hours through one prediction rather than iterative prediction.
Changes in manuscript: 'Also, the temporal resolution of satellite data is more coarse (30 minutes), so we can directly get the sequence of four frames in the following 2 hours through one prediction rather than iterative prediction. Optical flow can predict such short sequence quickly and shows great advantages in saving computing resources and avoiding error accumulation.'

22. L87-88 – Explain more what you mean by "increases level through recursive application" and justify your use of satellite data.
Response: We have explained in the revised manuscript. The justification of our use of satellite data is in the response of comment 21.
Changes in manuscript: 'Besides, the biggest drawback of convolution is that it smooths the characteristics of image, and the level of smoothness increases when applying convolution recursively in deep learning models. Therefore, to ease the smoothing of radar echo and preserve more details of precipitating systems, we decide to use the results of predicted satellite data by Optical flow in our model.'

23. L120-124 –There is no mention of how the model was trained and the results were validated. Ideally this information should be in the methods
Response: We add a new part 'Model description' that shows the architecture of each of our model, including parameters such as kernel size, padding, drop out, learning rate, optimizer, loss function etc. Also, another part 'Reference models' shows how we compare with other baseline models. 'Training and evaluating method of Multi-source Data Model (MSDM)' shows how we train and evaluate MSDM.
Changes in manuscript: We add three parts to explain how the model was trained and the results were validated.

24. L120 – It almost appears that the aim of the paper has not been clearly stated. You are using multiple sources of data and at the same time creating a multi-source data model (MSDM). This is very difficult to follow throughout the paper. Make this distinction clear

Response: We will explain our aim in the introduction part of the revised manuscript.
Changes in manuscript: 'On one hand, the current massive amounts of data are underutilized, on the other hand, scientists in the field of machine learning focus on pursuing high accuracy by increasing the complexity of models based on a single source of data. Given this background, from the perspective of atmospheric science, we build a multi-source data model (MSDM) with the aim to fully use multi-source observation data and find suitable machine learning algorithms for each type of data that can ensure accuracy while saving computing resources. Besides, due to the high degree of freedom and non-linearity of neural network, it is hard to apply physical constraints to theses machine learning models. Hence, we hope multi-source data will function as a proxy of physical constraints to guide the model in the training process.'

25. L125 – Use other metrices to evaluate model performance
Response: We introduce more metrics to evaluate model performances: CSI, HSS, FAR, RMSE, SSIM.
Changes in the manuscript: We add a new part 'Performance evaluation' to introduce the metrics we use to evaluate model performance. In the 'Results' part, the evaluation are shown in the form of table and graph.

26. L148 – Who made that claim? Citation ?
Response: Yu et.al (2018) made the claim that 'recurrent networks for sequence learning require iterative training, which introduces error accumulation by steps.'
Changes in manuscript: We add the citation. 'The ConvLSTM is prone to error accumulation due to the iterative training and needs massive computing resources (Yu et al., 2018).'

27. L195-210 - Include a thorough discussion to examine the relevance of your results and how it relates to other studies, previous methods used etc.
Response: We discuss the relevance of our models with other studies in the revised manuscript.
Changes in manuscript: In the 'Conclusions and discussions' part, we discuss the background of existing study and explain our contribution to this problem. We copy a small part here and the whole part has been revised in the manuscripts:
'As a conventional precipitation nowcasting method, Optical Flow has played a certain role in the forecast of advective precipitation. However, it performs poor on the prediction of advective precipitation due to the simplicity of its algorithm and the lack of use of existing massive data(Woo and Wong, 2017). Meanwhile, deep learning shows great advantages in processing vast amount of data. By using convolution and LSTM structures, deep learning algorithms are better at capturing spatiotemporal correlations. Nevertheless, the recurrent network (represented by ConvLSTM) for predicting spatial-temporal sequence are widely known to be difficult to train and computationally expensive (Yu et.al, 2018). Compared with the traditional spatial-temporal sequence task in the field of machine learning, such as moving mnist prediction, human position prediction, traffic flow prediction, REE task has its specific background and physical

constraints. Therefore, merely getting predictions with higher scores does not reflect the quality of the results. Wang et.al (2018,2019) designed state-of-art models to capture comprehensive correlations between spatial-temporal sequences. But when we apply them to the physics-based tasks represented by REE and QPN, we must evaluate its prediction from the point of atmospheric science. The prediction is of reference significance only when it is physically reasonable rather than with high scores. However, it is difficult to apply physical constraints to neural network due to its high degree of freedom and non-linearity. Hence, we input more kinds of data as features to the network with the intention that it could get more information through feature interaction. Therefore, we collect multi-source data and design MSDM. There could be a situation that when the model goes to a wrong way and tries to predict low radar reflectivity, the incorporated satellite data will balance it. We hope the multi-source data function as a proxy of constraint to the model. Solving sequence-to-sequence problem is computationally expensive, so we treat the QPN as an image-to-image problem and design MSDM based on a CNN (U-Net) with high efficiency and few parameters. The biggest advantage of MSDM is its transferability. Apart from satellite data, any other data (wind speed, pressure, temperature, etc.) can be used as input to the model in the future. Wind speed data could add dynamic constraints, and temperature data could add thermodynamic constraint…'

28. L195 – Can it be conclusively stated that looking at the problem through image-image prediction is better than focusing on the problem as spatio-temporal sequence problem. If that is the aim of your paper, then what is the conclusion? Deliberate further.

Response: We evaluate the four models in terms of 12 aspects to show the advantages and drawbacks of the models and of their combinations.

Changes in manuscript: We use a table to evaluate these models and discuss their advantages and drawbacks. The aim of our paper also be concluded in the 'Conclusions and dicussions'.

Reference:
Ho, J., Kalchbrenner, N., Weissenborn, D., and Salimans, T.: Axial Attention in Multidimensional Transformers, 2019.

Ronneberger, O., Fischer, P., and Brox, T.: U-Net: Convolutional Networks for Biomedical Image Segmentation, in: Medical Image Computing and Computer-Assisted Intervention – MICCAI 2015, vol. 9351, edited by: Navab, N., Hornegger, J., Wells, W. M., and Frangi, A. F., Springer International Publishing, Cham, 234–241, https://doi.org/10.1007/978-3-319-24574-4_28, 2015.

Yu, B., Yin, H., Zhu, Z.:Spatio-Temporal Graph Convolutional Networks: A Deep Learning Framework for Traffic Forecasting, 2018

---

## Author Comment (AC2) · 23 Apr 2021

1.  The critical success indexes (CSI) depends on the artificial threshold chosen by the authors, making it difficult to judge whether MSDM performs better than Optical flow and ConvLSTM. The authors might choose several thresholds (i. e., 0.1, 0.3, 1, 3, 10, 40), and calculate the average CSI of these thresholds, then we can compare these models more easily

Response: Thank you for your advice. We choose six thresholds (0.1, 1, 5, 10, 25, 40) and introduce more metrics (HSS,FAR,SSIM) to evaluate model performances. To stress the importance of areas with large radar reflectivity, we assign a weight w( threshold ) (Eq. 9) to different thresholds and calculate the weighted CSI and HSS Changes in manuscript:

$$w(\text{threshold}) = \begin{cases} 1, & \text{threshold} = 0.1 \\ 1, & \text{threshold} = 1 \\ 2, & \text{threshold} = 5 \\ 3, & \text{threshold} = 10 \\ 5, & \text{threshold} = 25 \\ 8, & \text{threshold} = 40 \end{cases} \tag{9}$$

**Table 1.** Weighted average CSI on test set with different thresholds (0.1, 1, 5, 10, 25, 40, unit: dBZ). The best score is in bold-face. The second-best score is underscored (The greater the better).

| Model | 30 min | 60 min | 90 min | 120 min |
|---|---|---|---|---|
| Optical Flow | **0.414** | 0.303 | 0.209 | 0.205 |
| ConvLSTM | 0.399 | 0.269 | 0.211 | 0.157 |
| U-Net | 0.348 | 0.259 | 0.216 | 0.184 |
| MSDM_mse | 0.362 | 0.286 | 0.245 | **0.218** |
| MSDM_ssim | 0.405 | **0.317** | **0.258** | 0.217 |

**Table 2.** Weighted average HSS on test set with different thresholds (0.1, 1, 5, 10, 25, 40, unit: dBZ). The best score is in bold-face. The second-best score is underscored (The greater the better).

| Model | 30 min | 60 min | 90 min | 120 min |
|---|---|---|---|---|
| Optical Flow | 0.512 | 0.409 | 0.34 | **0.304** |
| ConvLSTM | 0.487 | 0.311 | 0.246 | 0.18 |
| U-Net | 0423 | 0.307 | 0.25 | 0.209 |
| MSDM_mse | 0.437 | 0.341 | 0.29 | 0.255 |
| MSDM_ssim | **0.514** | **0.413** | **0.343** | 0.291 |

**Table 3.** Average FAR on test set with different thresholds (0.1, 1, 5, 10, 25, 40, unit: dBZ). The best score is in bold-face. The second-best score is underscored (The smaller the better).

| Model | 30 min | 60 min | 90 min | 120 min |
|---|---|---|---|---|
| Optical Flow | 0.316 | 0.391 | 0.439 | 0.474 |
| ConvLSTM | 0.265 | 0.295 | **0.242** | **0.246** |
| U-Net | 0.293 | 0.309 | 0.313 | 0.309 |
| MSDM_mse | 0.329 | 0.364 | 0.387 | 0.399 |
| MSDM_ssim | **0.237** | **0.27** | 0.303 | 0.335 |

2. RMSE is used frequently to judge the performance of machine learning models, but the RMSE of MSDM is too high compared with optical flow, so I suggest the authors to improve and re-train the MSDM model to get a lower RMSE.

Response: We modify the parameters of MSDM and get lower RMSE. Besides, we also trained MSDM with MSE loss function, which has the lowest RMSE. We show the comparison in the revised manuscripts. There are three points to be noted here: 1. MSDM was trained with SSIM loss function, whereas other models were trained with the MSE loss function. Therefore, MSDM is not going to minimize the RMSE but to maximize the structural similarity. 2. RMSE evaluate the global error of predictions. But it cannot evaluate the performance of local area. MSDM gets higher CSI and lower FAR, so it predicts better than other models in local area. 3. We should not focus on one metric. In the revised manuscripts, MSDM outperforms other models in SSIM, CSI, HSS and FAR.

Changes in manuscript: The results of MSDM trained with MSE loss function has been added in the revised manuscripts. It achieves lowest RMSE but performs poor on other metric.

[Figure]

3. The MSDM should be described in more details. In Fig. 3, the red arrow on "ours" (MSDM) indicates that the optical flow is used in MSDM, but in Fig. 4 there is no optical flow in the structure of MSDM?

Response: Fig 3 is a general description of the four models, including the iterative process after one round of predictions. Fig 4 provides more details about the deep learning part of MSDM (feature map, skip connection, convolution etc.). We will put every part of MSDM in Fig 4 in the revised manuscript (including Deep learning, Optical Flow, Random Forest).

Changes in manuscript: We modified Fig 4 and explain it in the new part 'Model architecture'.

[Figure]

4. Why not predict precipitation directly using MSDM?

Response: We explain the reason why we do not predict precipitation directly using MSDM in the 'Model architecture' part.

Changes in manuscript:

'The reason why we do not predict precipitation directly using deep learning part are as follows: 1) The precipitation data we collected is irregular site data, which is only distributed on land and does not include precipitation on the sea (Figure 1). Whereas the combined radar reflectivity (Figure 2a) and Himawari 8 satellite data (Figure 2b) are regular grid point data and include the data on the sea. The spatial distribution of these three types of data is inconsistent, so it is impossible to make a feature-label correspondence to directly predict precipitation. 2) The use of shapefiles to extract radar echo or satellite data on land will cause the edge of the echo to be limited to the land, which loses the meaning of extrapolation. 3) We hope to improve the transferability of MSDM that can integrate different kinds of data except grid point data. Therefore, the method of processing precipitation data can be used on other observation site data in the daily operation. 4) We believe that deep learning extracts the long-period trend of precipitation efficiently, but it cannot capture the transient characteristics of precipitation. Therefore, for each rainfall event, we use Random forest to model the non-linear relationship between multi-source data to capture its unique characteristics.'

---

## Author Comment (AC4) · 23 Apr 2021

1. L034 - What does the HRRR acronym stand for?
Response: High Resolution Rapid Refresh (HRRR). We have explained the acronyms in the revised manuscript.
Changes in manuscript:
'which is superior to High Resolution Rapid Refresh (HRRR) numerical prediction from National Oceanic and Atmospheric Administration (NOAA) when the prediction time is within 6 hours.'

2. L035 - Explain what are the "U-Net" and "Met-Net" methods.
Response: We have explained "U-Net" and "Met-Net" in the revised manuscript.
Changes in the manuscript:
'U-Net(Ronneberger et al., 2015) is a well-known network designed for image segmentation, and its core is up-sampling, down-sampling and skip connection. It can efficiently achieve high accuracy with a small number of samples.'
'Sonderby et al. (2020) proposed a Neural Weather Model (NWM) called MetNet that uses axis self-attention (Ho et al.,2019) to discover the weather pattern from radar and satellite data. MetNet can predict the next 8 hours precipitation with a resolution of 1 kilometer in 2-minute intervals.'

3. L038 - Explain what is a TrajGRU model.
Response: We have explained TrajGRU in the revised manuscript.
Changes in the manuscript: 'Furthermore, they apply the same modification to Gated Recurrent Unit (GRU), and notice that convolution is location-invariant and only focus on fixed location because its hyperparameter (kernel size, padding, dilation) is fixed. But in the QPN problem, a specific location of cloud clusters continuously changes over time. Hence, Shi et al. (2016) proposed Trajectory Gated Recurrent Unit (TrajGRU) that use a subnetwork to output a location-variant connection structure before state transitions. The dynamically changed connections help TrajGRU to capture the trajectory of cloud clusters more accurately. '

4. L040 - Some concise comments about the PredRNN++, MIM, and E3D-LSTM networks are necessary.
Response: We elaborate the description of PredRNN++, MIM, and E3D-LSTM networks.
Changes in the manuscript: 'In the field of video prediction, Wang et al. proposed various recurrent networks based on LSTM. For example, they designed PredRNN++ (Wang et al., 2018) with cascaded dual memory structure and gradient highway unit, which strengthens the power for modelling short-term dynamics and alleviates the vanishing gradient problem respectively. In addition, to capture spatial characteristics through the recurrent state transitions, Wang et.al(2019a) integrate the 3D convolution inside the LSTM units and proposed Eidetic 3D LSTM(E3D-LSTM). Moreover, Wang et al. (2019b) designed Memory in Memory (MIM) to handle higher-order non-stationarity of spatio-temporal data. By using differential signals, MIM can model the non-stationary properties between adjacent recurrent states. However, their work is

based on a slight modification of existing techniques demanding massive computing resource to train and have not been applied to the numerous meteorological data.'

5. L065 - The Figure 1 is not legible. You should improve it.
Response: We have improved it.
Changes in the manuscript: We replace Figure 1 with a legible graph.

[Figure]

6. L122 - I wonder if 240 days of data are enough to train the MSDM. Is this choice explained by a limitation in the computations or is there another justification?
Response: Yes, 240 days may not be enough for training due to the limitation of collecting data. But we make the following justification:
1) We collect 292 days of data and split them into three parts: 80% for training set, 10% for validation set, and 10% for test set. The training set includes several typical types of rainfall events over East China: Convective precipitation, Advection precipitation, Typhoon precipitation.
2) The larger amount of data is to prevent overfitting of the model and enhance the generalization ability of the model. We introduce the early-stopping strategy to monitor the model's performance on validation set to prevent overfitting.
3) U-Net has been proved that it can achieve high accuracy on small number of samples. Therefore, we believe the deep learning part of MSDM based on the modification of U-Net has the same ability.
4) We think that the characteristics of precipitation in a region keep changing over time. The model we trained is based on the data of recent years. Hence it could capture the recent characteristics of the precipitation. Training with long-term data will obtain more general characteristics, while erasing these typical unique characteristics.
5) In the future, we will collect more data to do further research.

7. L 135 - Figure 5 corresponds to a particular date and time. The authors should indicate what they are on the figure. Moreover, I wonder what would be the results for other dates and times. There are too few results presented for the validation and test of the AI methods. More results should be shown.

Response: The date and time of figure 5 is 201809070000. We will add other examples in the revised manuscript.

Changes in manuscript: We add the date and time of figure 5. More examples will be shown in the revised manuscript.

8. L 142 - I do not understand: "it tracks features by the corner detector". What does it mean?

Response: In computer vision, corner (also known as interest points) is the uniquely recognizable characteristics of a image. Corner detector, for example, Harris corner detector, is one of the algorithms for searching these corners. More details will be found at https://docs.opencv.org/3.1.0/d4/d7d/tutorial_harris_detector.html. We will rephrase the sentence to make it easy to comprehend.

Changes in the manuscript: 'However, the fatal weakness of the Optical flow method is that it simply predicts the movement of radar echo from previous images without predicting decay and initiation of radar echo, which causes its accuracy to decrease over time (Table 1) and the false alarm ratio keeps increasing (Table 3). Besides, it employs an algorithm called corner detector (Ayzel et al., 2019) to identify special points from previous frames, and track the movement of these points. When it extrapolates the tail of radar echo, it cannot find corresponding points from previous images (due to the tail of the radar echo at this moment was in a position outside the radar image of previous frames). Consequently, there exist unreasonable shapes in the tail of predicted radar echo.'

9. L 156 - Table 1 - I guess that the Critical Sucess Index is given for four methods, but only for one date and time. What about other test-cases? I think that the methods should be benchmarked in a large number of situations in order to be able to comment the scores.

Response: Table 1 is the average CSI on test set of four models, not a certain day. In the revised manuscript, we choose six thresholds (0.1, 1, 5, 10, 25, 40) and introduce more metrics (HSS,FAR,SSIM) to evaluate model performances.

Changes in the manuscript:

**Table 1.** Weighted average CSI on test set with different thresholds (0.1, 1, 5, 10, 25, 40, unit: dBZ). The best score is in bold-face. The second-best score is underscored (The greater the better).

| Model | 30 min | 60 min | 90 min | 120 min |
|---|---|---|---|---|
| Optical Flow | **0.414** | 0.303 | 0.209 | 0.205 |
| ConvLSTM | 0.399 | 0.269 | 0.211 | 0.157 |
| U-Net | 0.348 | 0.259 | 0.216 | 0.184 |
| MSDM_mse | 0.362 | 0.286 | 0.245 | **0.218** |
| MSDM_ssim | 0.405 | **0.317** | **0.258** | 0.217 |

**Table 2.** Weighted average HSS on test set with different thresholds (0.1, 1, 5, 10, 25, 40, unit: dBZ). The best score is in bold-face. The second-best score is underscored (The greater the better).

| Model | 30 min | 60 min | 90 min | 120 min |
|---|---|---|---|---|
| Optical Flow | 0.512 | 0.409 | 0.34 | **0.304** |
| ConvLSTM | 0.487 | 0.311 | 0.246 | 0.18 |
| U-Net | 0423 | 0.307 | 0.25 | 0.209 |
| MSDM_mse | 0.437 | 0.341 | 0.29 | 0.255 |
| MSDM_ssim | **0.514** | **0.413** | **0.343** | 0.291 |

**Table 3.** Average FAR on test set with different thresholds (0.1, 1, 5, 10, 25, 40, unit: dBZ). The best score is in bold-face. The second-best score is underscored (The smaller the better).

| Model | 30 min | 60 min | 90 min | 120 min |
|---|---|---|---|---|
| Optical Flow | 0.316 | 0.391 | 0.439 | 0.474 |
| ConvLSTM | 0.265 | 0.295 | **0.242** | **0.246** |
| U-Net | 0.293 | 0.309 | 0.313 | 0.309 |
| MSDM_mse | 0.329 | 0.364 | 0.387 | 0.399 |
| MSDM_ssim | **0.237** | **0.27** | 0.303 | 0.335 |

10. Otherwise, the MSDM ranks very differently depending on the observation times (from 30 to 120 minutes) with the 0.1 dBZ threshold. Is it logical and explainable? Is it worth noticing that the MSDM ranks consistently (second best score) with the 40 dBZ thresholds. What would be the scores of the MSDM for the Radar Echo Extrapolation at other dates and times?

Response: We choose six thresholds (0.1, 1, 5, 10, 25, 40) and introduce more metrics (HSS,FAR,SSIM) to evaluate model performances. To stress the importance of areas with large radar reflectivity, we assign a weight w( threshold ) (Eq. 9) to different thresholds and calculate the weighted CSI and HSS.

Changes in the manuscript:

$$w(\text{threshold}) = \begin{cases} 1, & \text{threshold} = 0.1 \\ 1, & \text{threshold} = 1 \\ 2, & \text{threshold} = 5 \\ 3, & \text{threshold} = 10 \\ 5, & \text{threshold} = 25 \\ 8, & \text{threshold} = 40 \end{cases} \tag{9}$$

The results of scores have been shown in the previous comment.

11. L 165 -It seems that using the Modified Structural Similarity Index (denoted by SSIM) is counter-productive in terms of MAE and RMSE. Why use it? Once more, I wonder if the example of results produced for one date and time has a general value.

Response: We train each model with MSE loss function to make comparison with the model trained with SSIM. Examples and evaluating scores have been shown in the previous response. Fig 5 shows that MSE cannot predict the large-value area of radar echo.

Changes in the manuscript: We train the MSDM using MSE loss function to make comparison with the model trained with SSIM (Fig 1 and Fig 2). Explanations have been made in the revised manuscript.

[Figure]

Figure 1 Models trained with SSIM

Figure 2 Models trained with MSE

12. L 186 - Figure 8 - Looking at this figure, I am not very convinced that the CSI of the Quantitative Precipitation Nowcasting are better using the random forest than using the Z-R relationship. In general, the scores are quite similar. Could the authors try to better advocate the random forest method?

Response: The CSI describe the spatial distribution of precipitation. We use RMSE and to describe the accuracy of different methods in the revised manuscript. When estimating the precipitation rate at a specific period, the number of data sample is very small, so it is not suitable for methods such as deep learning that require big data. Regressive method such as Random forest will perform better on a small sample of data.

Changes in the manuscript:

[Figure]

13. L 212 - The acronyms "RNN" and "GRU" should be developed.

Response: It has been developed.

Changes in the manuscript: 'Recurrent Neural Network(RNN), Gated Recurrent Unit(GRU).'

14. My general feeling about the AI methods used separately or combined together through the paper is that all of them have advantages and drawbacks. I suggest to the authors to add a final synthetic table describing the strong points and weak points of the methods and of their combinations. This would greatly help the readers to understand the arguments of the authors.

Response: Thank you very much for this suggestion. We summarize these methods and evaluate them in terms of 12 aspects in the revised manuscript.

Changes in the manuscript: We copy the Table 4 here and the discussion about their advantages and drawbacks has been made in the revised manuscripts.

**Table 4.** Evaluation on four models with (The less the better)

| | The amount of data required for training↓ | Time used for training model↓ | False Alarm Rate↓ | Accumulative system error↓ |
|---|---|---|---|---|
| Optical flow | 1 | 1 | 2 | 1 |
| ConvLSTM | 4 | 4 | 3 | 2 |
| U-Net | 2 | 2 | 4 | 2 |
| MSDM | 3 | 3 | 1 | 4 |

(The more the better)

| | The ability to capture spatial characteristics↑ | The ability to capture temporal characteristics↑ | The ability to predict initiation and decay of radar echo↑ | 0~1 hour forecast accuracy↑ |
|---|---|---|---|---|
| Optical flow | 1 | 3 | 1 | 3 |
| ConvLSTM | 2 | 4 | 2 | 1 |
| U-Net | 3 | 1 | 3 | 2 |
| MSDM | 4 | 1 | 4 | 4 |

(The more the better)

| | 1~2 hour forecast accuracy↑ | The ability to maintain the shape of radar echo↑ | Clarity of radar image↑ | Conform to the laws of physics↑ |
|---|---|---|---|---|
| Optical flow | 1 | 4 | 4 | 4 |
| ConvLSTM | 4 | 1 | 1 | 1 |
| U-Net | 3 | 2 | 2 | 2 |
| MSDM | 2 | 3 | 3 | 3 |

---

## Author Response (AR1)

Referee comment 1:
1. L9 – Change to 'predict precisely'
Response: The change has been made.
Changes in the manuscript: 'in the imminent future the rainfall rate affected by which is difficult to precisely predict precisely'

2. L13 – insert 'is' between that and suitable
Response: The change has been made.
Changes in the manuscript: 'it is important to train a data-driven model from scratch that is suitable to…'

3. L13 – change collect to collected
Response: The change has been made.
Changes in the manuscript: 'We collected three kinds of data (radar, satellite, precipitation) in flood season…'

4. L22 – Define SSIM
Response: We develop the acronyms SSIM as Structural Similarity (SSIM) and the definition of which is in eq.4:

$$Loss = -1 \times SSIM(y_{pred}, y_{true}) = -1 \times \frac{\left(2\mu_{y_{pred}}\mu_{y_{true}} + C_1\right)\left(2\sigma_{y_{pred}y_{true}} + C_2\right)}{\left(\mu_{y_{pred}}^2 + \mu_{y_{true}}^2 + C_1\right)\left(\sigma_{y_{pred}}^2 + \sigma_{y_{true}}^2 + C_2\right)}$$

Changes in the manuscript: 'we applied a modified Structural Similarity (SSIM) index as a loss function.'

5. L23 – explain the traditional Z-R relationship
Response: The change has been made.
Changes in the manuscript: 'the results outperform those of the traditional Z-R relationships that use logarithmic function to describe the non-linear relationships between radar reflectivity and rainfall rate'

6. What is NOAA's HRRR? There is a need to go through the entire paper and define all the acronyms in its first use, even for something as common as AI.
Response: National Oceanic and Atmospheric Administration (NOAA) High Resolution Rapid Refresh (HRRR). Other acronyms have been defined.
Changes in the manuscript: 'which is superior to High Resolution Rapid Refresh (HRRR) numerical prediction from the National Oceanic and Atmospheric Administration (NOAA) when the prediction time is within 6 hours.'

7. L35 – Remove repeated citation
Response: The change has been made.
Changes in the manuscript: 'Sonderby et al.(2020) proposed'

8. L35 – What is MetNet? Do not assume the reader knows

Response: We have explained the 'MetNet' in the revised manuscript.

Changes in the manuscript: 'Sonderby et al.(2020) proposed a neural weather model (NWM) called MetNet that uses axis self-attention (Ho et al., 2019) to discover weather patterns from radar and satellite data. MetNet can predict the next 8 hours of precipitation in 2-minute intervals with a resolution of 1 kilometer.'

9.  L37 – Remove repeated citation and do this for every other instance where this is the case

Response: The change has been made and the entire paper has been thoroughly edited for English writing and grammar.

Changes in the manuscript: 'Shi et al.(2015)'

10. L37 – How were Shi.et al able to achieve this? Explain their method of prediction of spatio-temporal predictions

Response: We have explained their method in the revised manuscript.

Changes in the manuscript: 'Shi et al. (2015) treated precipitation nowcasting as a problem of predicting spatiotemporal sequences and modified the fully connected long short-term memory (FC-LSTM) by replacing the Hadamard product with a convolution operation in the input-to-state and state-to-state transitions. They believe that cloud movement is highly uniform in some areas, and convolutions can capture these local characteristics. Therefore, the convolution operation in the input transformations and recurrent transformations of their proposed convolutional LSTM (ConvLSTM) helps to handle the spatial correlations. Furthermore, they apply the same modification to the gated recurrent unit (GRU) and notice that convolution is location-invariant and focuses on only a fixed location because its hyperparameters (kernel size, padding, dilation) are fixed. However, in the QPN problem, a specific location of cloud clusters continuously changes over time. Hence, Shi et al. (2017) proposed a trajectory GRU (TrajGRU) that uses a subnetwork to output a location-variant connection structure before state transitions. The dynamically changed connections help TrajGRU capture the trajectory of cloud clusters more accurately than previous methods.'

11. L41 – Avoid the use of contractions eg. Haven't

Response: The change has been made.

Changes in the manuscript: 'has not been applied to big meteorological data'

12. L43 – Avoid using etc. if you cannot name more items

Response: The change has been made.

Changes in the manuscript: 'Computer vision techniques have long been used in object detection, video prediction, and human motion prediction.'

13. L44 – L45 – incorrect use of tense, past (used) and present (will mislead). Check for grammatical errors throughout the paper

Response: The use of tense has been corrected and the entire paper has been proof read by an English first language checker.

Changes in the manuscript: 'Song (2019) used image quality assessment techniques as a new loss function instead of the common mean squared error (MSE), which misled the process of training and generated blurry images.'

14. L45 – How is optical flow method related to Trans method? Or is it not related? Why is it mentioned.

Response: It is related to Ayzel et al 's work and has been moved to the right part.

Changes in the manuscript: 'Ayzel et al. (2019) designed an advanced model based on the multiple optical flow algorithm for QPN, but it still performs poorly in the prediction of the onset and decay of precipitation systems because optical flow methods simply calculate the position and velocity of the radar echo with a constant velocity rather than consider the changing intensity of radar echo.'

15. L51 – Remove et.al

Response: We have removed 'et.al'

Changes in the manuscript: 'Given this background, from the perspective of atmospheric science, we build a multisource data model (MSDM)…'

16. L59 – Rephrase "to train the deep learning model to learn"

Response: It has been rephrased.

Changes in the manuscript: ' To train a deep learning model that can capture the precipitation characteristics of East China'.

17. L66 – Provide links for datasets/sources

Response: The links have been provided in the revised manuscript.

Changes in the manuscript:

'Radar data: http://data.cma.cn/data/detail/dataCode/J.0012.0003.html , AWS data: http://data.cma.cn/data/detail/dataCode/A.0012.0001.html , Himawari 8 satellite data: http://www.cr.chiba-u.jp/databases/GEO/H8_9/FD/index.html '

18. L74 – change "we compared" to a comparison was made

Response: The change has been made.

Changes in the manuscript: 'To test our method, comparison was made…'

19. L75 – What exactly is being predicted first? Needs more clarity

Response: We have clarified in the revised manuscript.

Changes in the manuscript: 'Due to limits on computational resource, we use a few frames to predict the results for the half-hour. Then, the output results are used to iteratively predict the radar echo in the next half-hour to achieve a lead time of 2 hours (Fig 4). For the baseline sequence-to-sequence models (ConvLSTM, Optical flow), we use the first 5 frames ($T_{-4} \sim T_0$) to predict a sequence of the next 5 frames($T_1 \sim T_5$), and use this result to iteratively predict the remaining three sequences ($T_6 \sim T_{10}$, $T_{11} \sim T_{15}$, $T_{16} \sim T_{20}$). For image-to-image models (U-Net, MSDM), we use frame $T_0$ to predict frame $T_5$, and use this prediction as input to iteratively predict the following frames ($T_{10}$,

$T_{15}$, $T_{20}$).'

20. L84 – change to "makes it better to predict". The entire paper needs to be thoroughly edited for English writing and grammar.
Response: The change has been made and the entire paper has been thoroughly edited for English writing and grammar.
Changes in manuscript: 'make it better to predict…'

21. L84 – Are you trying to say that the satellite data is more coarse ? Or has low spatial resolution? Or are you referring to the time resolution?
Response: Yes, the temporal resolution of satellite data is more coarse. Its interval is 30 minutes. We try to express that it only has four frames in the lead time of 2 hours. Therefore, we can get the sequence of the following 2 hours through one prediction rather than iterative prediction.
Changes in manuscript: 'Additionally, the temporal resolution of satellite data is coarser (30 minutes), so we can directly obtain the sequence of four frames of the following 2 hours through one prediction rather than iterative prediction. Optical flow can predict such short sequences quickly and shows great advantages in saving computing resources and avoiding error accumulation.'

22. L87-88 – Explain more what you mean by "increases level through recursive application" and justify your use of satellite data.
Response: We have explained in the revised manuscript. The justification of our use of satellite data is in the response of comment 21.
Changes in manuscript: 'In addition, the main drawback of the convolution operation is that it smooths the characteristics of the image, and the level of smoothness increases when applying convolutions recursively in deep learning models. Therefore, to ease the smoothing of radar echoes and preserve more details of precipitation systems, we decide to use the results of satellite data predicted by the optical flow component of our model.'

23. L120-124 –There is no mention of how the model was trained and the results were validated. Ideally this information should be in the methods
Response: We add a new part 'Model description' that shows the architecture of each of our model, including parameters such as kernel size, padding, drop out, learning rate, optimizer, loss function etc. Also, another part 'Reference models' shows how we compare with other baseline models. 'Training and evaluating method of Multi-source Data Model (MSDM)' shows how we train and evaluate MSDM.
Changes in manuscript: We add three parts to explain how the model was trained and the results were validated.

24. L120 – It almost appears that the aim of the paper has not been clearly stated. You are using multiple sources of data and at the same time creating a multi-source data model (MSDM). This is very difficult to follow throughout the paper. Make this

distinction clear

Response: We will explain our aim in the introduction part of the revised manuscript.

Changes in manuscript: 'On the one hand, the current massive amounts of data are underutilized; on the other hand, scientists in the field of machine learning focus on pursuing high accuracy by increasing the complexity of models based on a single source of data. Given this background, from the perspective of atmospheric science, we build a multisource data model (MSDM) with the aim of fully using multisource observation data (for example, radar reflectivity, infrared satellite data, and rain gauge data) and find suitable machine learning algorithms (for example, deep neural network, optical flow, and random forest algorithms) for each type of data that can ensure accuracy while saving computing resources. In addition, due to the high degrees of freedom and nonlinearity of neural networks, it is difficult to apply physical constraints to these machine learning models. Hence, we hope that multisource data will function as a proxy for physical constraints to guide the model during the training process.'

25. L125 – Use other metrices to evaluate model performance

Response: We introduce more metrics to evaluate model performances: CSI, HSS, FAR, RMSE, SSIM.

Changes in the manuscript: We add a new part 'Performance Evaluation' to introduce the metrics we use to evaluate model performance. In the 'Results' part, the evaluations are shown in the form of table and graph.

26. L148 – Who made that claim? Citation ?

Response: Yu et.al (2018) made the claim that 'recurrent networks for sequence learning require iterative training, which introduces error accumulation by steps.'

Changes in manuscript: We add the citation. 'ConvLSTM is prone to error accumulation due to iterative training and requires massive computing resources (Yu et al., 2018). '

27. L195-210 - Include a thorough discussion to examine the relevance of your results and how it relates to other studies, previous methods used etc.

Response: We discuss the relevance of our models with other studies in the revised manuscript.

Changes in manuscript: In the 'Conclusions and discussions' part, we discuss the background of existing study and explain our contribution to this problem.

28. L195 – Can it be conclusively stated that looking at the problem through image-image prediction is better than focusing on the problem as spatio-temporal sequence problem. If that is the aim of your paper, then what is the conclusion? Deliberate further.

Response: We evaluate the four models in terms of 12 aspects to show the advantages and drawbacks of the models and of their combinations.

Changes in manuscript: We use a table to evaluate these models and discuss their advantages and drawbacks. The aim of our paper also be concluded in the 'Conclusions and dicussions'.

Reference:

Ho, J., Kalchbrenner, N., Weissenborn, D., and Salimans, T.: Axial Attention in Multidimensional Transformers, 2019.

Ronneberger, O., Fischer, P., and Brox, T.: U-Net: Convolutional Networks for Biomedical Image Segmentation, in: Medical Image Computing and Computer-Assisted Intervention – MICCAI 2015, vol. 9351, edited by: Navab, N., Hornegger, J., Wells, W. M., and Frangi, A. F., Springer International Publishing, Cham, 234–241, https://doi.org/10.1007/978-3-319-24574-4_28, 2015.

Yu, B., Yin, H., Zhu, Z.:Spatio-Temporal Graph Convolutional Networks: A Deep Learning Framework for Traffic Forecasting, 2018

Referee comment 2:

1. The critical success indexes (CSI) depends on the artificial threshold chosen by the authors, making it difficult to judge whether MSDM performs better than Optical flow and ConvLSTM. The authors might choose several thresholds (i. e., 0.1, 0.3, 1, 3, 10, 40), and calculate the average CSI of these thresholds, then we can compare these models more easily

Response: Thank you for your advice. We choose six thresholds (0.1, 1, 5, 10, 25, 40) and introduce more metrics (HSS,FAR,SSIM) to evaluate model performances. To stress the importance of areas with large radar reflectivity, we assign a weight w( threshold ) (Eq. 9) to different thresholds and calculate the weighted CSI and HSS Changes in manuscript:

$$w(\text{threshold}) = \begin{cases} 1, & \text{threshold} = 0.1 \\ 1, & \text{threshold} = 1 \\ 2, & \text{threshold} = 5 \\ 3, & \text{threshold} = 10 \\ 5, & \text{threshold} = 25 \\ 8, & \text{threshold} = 40 \end{cases} \tag{9}$$

**Table 1.** Weighted average CSI on test set with different thresholds (0.1, 1, 5, 10, 25, 40, unit: dBZ). The best score is in bold-face. The second-best score is underscored (The greater the better).

| Model | 30 min | 60 min | 90 min | 120 min |
| --- | --- | --- | --- | --- |
| Optical Flow | **0.414** | 0.303 | 0.209 | 0.205 |
| ConvLSTM | 0.399 | 0.269 | 0.211 | 0.157 |
| U-Net | 0.348 | 0.259 | 0.216 | 0.184 |
| MSDM_mse | 0.362 | 0.286 | 0.245 | **0.218** |
| MSDM_ssim | 0.405 | **0.317** | **0.258** | 0.217 |

**Table 2.** Weighted average HSS on test set with different thresholds (0.1, 1, 5, 10, 25, 40, unit: dBZ). The best score is in bold-face. The second-best score is underscored (The greater the better).

| Model | 30 min | 60 min | 90 min | 120 min |
| --- | --- | --- | --- | --- |
| Optical Flow | 0.512 | 0.409 | 0.34 | **0.304** |
| ConvLSTM | 0.487 | 0.311 | 0.246 | 0.18 |
| U-Net | 0423 | 0.307 | 0.25 | 0.209 |
| MSDM_mse | 0.437 | 0.341 | 0.29 | 0.255 |
| MSDM_ssim | **0.514** | **0.413** | **0.343** | 0.291 |

**Table 3.** Average FAR on test set with different thresholds (0.1, 1, 5, 10, 25, 40, unit: dBZ). The best score is in bold-face. The second-best score is underscored (The smaller the better).

| Model | 30 min | 60 min | 90 min | 120 min |
|---|---|---|---|---|
| Optical Flow | 0.316 | 0.391 | 0.439 | 0.474 |
| ConvLSTM | 0.265 | 0.295 | **0.242** | **0.246** |
| U-Net | 0.293 | 0.309 | 0.313 | 0.309 |
| MSDM_mse | 0.329 | 0.364 | 0.387 | 0.399 |
| MSDM_ssim | **0.237** | **0.27** | 0.303 | 0.335 |

2.  RMSE is used frequently to judge the performance of machine learning models, but the RMSE of MSDM is too high compared with optical flow, so I suggest the authors to improve and re-train the MSDM model to get a lower RMSE.

Response: We modify the parameters of MSDM and get lower RMSE. Besides, we also trained MSDM with MSE loss function, which has the lowest RMSE. We show the comparison in the revised manuscripts. There are three points to be noted here: 1. MSDM was trained with SSIM loss function, whereas other models were trained with the MSE loss function. Therefore, MSDM is not going to minimize the RMSE but to maximize the structural similarity. 2. RMSE evaluate the global error of predictions. But it cannot evaluate the performance of local area. MSDM gets higher CSI and lower FAR, so it predicts better than other models in local area. 3. We should not focus on one metric. In the revised manuscripts, MSDM outperforms other models in SSIM, CSI, HSS and FAR.

Changes in manuscript: The results of MSDM trained with MSE loss function has been added in the revised manuscripts. It achieves lowest RMSE but performs poor on other metric.

[Figure]

3. The MSDM should be described in more details. In Fig. 3, the red arrow on "ours" (MSDM) indicates that the optical flow is used in MSDM, but in Fig. 4 there is no optical flow in the structure of MSDM?

Response: Fig 3 is a general description of the four models, including the iterative process after one round of predictions. Fig 4 provides more details about the deep learning part of MSDM (feature map, skip connection, convolution etc.). We will put every part of MSDM in Fig 4 in the revised manuscript (including Deep learning, Optical Flow, Random Forest).

Changes in manuscript: Fig 4 and its explanation.

4. Why not predict precipitation directly using MSDM?

Response: We explain the reason why we do not predict precipitation directly using MSDM in the 'Model architecture' part.

Changes in manuscript:

'The reasons why we do not predict precipitation directly using deep learning are as follows: 1) The precipitation data we collected are irregular site data, which are distributed only on land and do not include precipitation on the sea (Fig 1). The combined radar reflectivity (Fig 2(a)) and Himawari 8 satellite data (Fig 2(b)) are regular grid point data and include sea data. The spatial distributions of these three types of data are inconsistent, so it is impossible to make a feature-label correspondence to directly predict precipitation. 2) The use of shapefiles to extract radar echo or satellite data on land will cause the edge of the echo to be limited to the land, which loses the meaning of extrapolation. 3) We hope to improve the transferability of MSDM that can integrate different kinds of data except grid point data. Therefore, the method of processing precipitation data can be used on other observation site data in daily operation. 4) We believe that deep learning efficiently extracts the long-period trend in precipitation, but it cannot capture the transient characteristics of precipitation. Therefore, for each rainfall event, we use random forest to model the nonlinear relationship between multisource data to capture its unique characteristics.'

Referee comment 3:

1.  L034 - What does the HRRR acronym stand for?

Response: High Resolution Rapid Refresh (HRRR). We have explained the acronyms in the revised manuscript.

Changes in manuscript:

'which is superior to High Resolution Rapid Refresh (HRRR) numerical prediction from the National Oceanic and Atmospheric Administration (NOAA) when the prediction time is within 6 hours.'

2.  L035 - Explain what are the "U-Net" and "Met-Net" methods.

Response: We have explained "U-Net" and "Met-Net" in the revised manuscript.

Changes in the manuscript:

'U-Net (Ronneberger et al., 2015) is a well-known network designed for image segmentation, and its core is upsampling, downsampling, and skip connection. It can efficiently achieve high accuracy with a small number of samples.'

'Sonderby et al.(2020) proposed a neural weather model (NWM) called MetNet that uses axis self-attention (Ho et al., 2019) to discover weather patterns from radar and satellite data. MetNet can predict the next 8 hours of precipitation in 2-minute intervals with a resolution of 1 kilometer.'

3.  L038 - Explain what is a TrajGRU model.

Response: We have explained TrajGRU in the revised manuscript.

Changes in the manuscript: ' Furthermore, they apply the same modification to the gated recurrent unit (GRU) and notice that convolution is location-invariant and focuses on only a fixed location because its hyperparameters (kernel size, padding, dilation) are fixed. However, in the QPN problem, a specific location of cloud clusters continuously changes over time. Hence, Shi et al. (2017) proposed a trajectory GRU (TrajGRU) that uses a subnetwork to output a location-variant connection structure before state transitions. The dynamically changed connections help TrajGRU capture the trajectory of cloud clusters more accurately than previous methods. '

4.  L040 - Some concise comments about the PredRNN++, MIM, and E3D-LSTM networks are necessary.

Response: We elaborate the description of PredRNN++, MIM, and E3D-LSTM networks.

Changes in the manuscript: 'In the field of video prediction, Wang et al. proposed various recurrent neural networks (RNNs) based on LSTM. For example, they designed PredRNN++ (Wang et al., 2018) with a cascaded dual memory structure and gradient highway unit, which strengthens the power for modeling short-term dynamics and alleviates the vanishing gradient problem, respectively. In addition, to capture spatial characteristics through recurrent state transitions, Wang et al. (2019a) integrated 3D convolutions inside LSTM units and proposed Eidetic 3D LSTM (E3D-LSTM). Moreover, Wang et al. (2019b) designed the memory in memory (MIM) network to handle higher-order nonstationarity of spatiotemporal data. By using differential signals,

MIM can model the nonstationary properties between adjacent recurrent states. However, their work is based on a slight modification of existing techniques demanding massive computing resources for model training and has not been applied to big meteorological data.'

5. L065 - The Figure 1 is not legible. You should improve it.
Response: We have improved it.
Changes in the manuscript: We replace Figure 1 with a legible graph.

[Figure]

6. L122 - I wonder if 240 days of data are enough to train the MSDM. Is this choice explained by a limitation in the computations or is there another justification?
Response: Yes, 240 days may not be enough for training due to the limitation of collecting data. But we make the following justification:
1) We collect 292 days of data and split them into three parts: 80% for training set, 10% for validation set, and 10% for test set. The training set includes several typical types of rainfall events over East China: Convective precipitation, Advection precipitation, Typhoon precipitation.
2) The larger amount of data is to prevent overfitting of the model and enhance the generalization ability of the model. We introduce the early-stopping strategy to monitor the model's performance on validation set to prevent overfitting.
3) U-Net has been proved that it can achieve high accuracy on small number of samples. Therefore, we believe the deep learning part of MSDM based on the modification of U-Net has the same ability.
4) We think that the characteristics of precipitation in a region keep changing over time. The model we trained is based on the data of recent years. Hence it could capture the recent characteristics of the precipitation. Training with long-term data will obtain more general characteristics, while erasing these typical unique

characteristics.

5) In the future, we will collect more data to do further research.

7. L 135 - Figure 5 corresponds to a particular date and time. The authors should indicate what they are on the figure. Moreover, I wonder what would be the results for other dates and times. There are too few results presented for the validation and test of the AI methods. More results should be shown.

Response: The date and time of figure 5 is 201809070000. We will add other examples in the revised manuscript.

Changes in manuscript: We add the date and time of figure 5. More examples will be shown in .

8. L 142 - I do not understand: "it tracks features by the corner detector". What does it mean?

Response: In computer vision, corner (also known as interest points) is the uniquely recognizable characteristics of a image. Corner detector, for example, Harris corner detector, is one of the algorithms for searching these corners. More details will be found at https://docs.opencv.org/3.1.0/d4/d7d/tutorial_harris_detector.html. We will rephrase the sentence to make it easy to comprehend.

Changes in the manuscript: 'However, the fatal weakness of the optical flow method is that it simply predicts radar echo movement from previous images without predicting radar echo decay and initiation, which causes its accuracy to decrease over time (Table 1), and the FAR keeps increasing (Table 3). In addition, it employs an algorithm called a corner detector (Ayzel et al., 2019) to identify special points from previous frames and track the movement of these points. When it extrapolates the tail of the radar echo, it cannot find corresponding points from previous images because the tail of the radar echo at this moment was in a position outside the radar image of previous frames. Consequently, unreasonable shapes exist in the tail of the predicted radar echo.'

9. L 156 - Table 1 - I guess that the Critical Sucess Index is given for four methods, but only for one date and time. What about other test-cases? I think that the methods should be benchmarked in a large number of situations in order to be able to comment the scores.

Response: Table 1 is the average CSI on test set of four models, not a certain day. In the revised manuscript, we choose six thresholds (0.1, 1, 5, 10, 25, 40) and introduce more metrics (HSS,FAR,SSIM) to evaluate model performances.

Changes in the manuscript:

**Table 1.** Weighted average CSI on test set with different thresholds (0.1, 1, 5, 10, 25, 40, unit: dBZ). The best score is in bold-face. The second-best score is underscored (The greater the better).

| Model | 30 min | 60 min | 90 min | 120 min |
|---|---|---|---|---|
| Optical Flow | **0.414** | 0.303 | 0.209 | 0.205 |
| ConvLSTM | 0.399 | 0.269 | 0.211 | 0.157 |
| U-Net | 0.348 | 0.259 | 0.216 | 0.184 |
| MSDM_mse | 0.362 | 0.286 | 0.245 | **0.218** |

| | | | |
|---|---|---|---|
| MSDM_ssim | 0.405 | **0.317** | **0.258** | 0.217 |

**Table 2.** Weighted average HSS on test set with different thresholds (0.1, 1, 5, 10, 25, 40, unit: dBZ). The best score is in bold-face. The second-best score is underscored (The greater the better).

| Model | 30 min | 60 min | 90 min | 120 min |
|---|---|---|---|---|
| Optical Flow | 0.512 | 0.409 | 0.34 | **0.304** |
| ConvLSTM | 0.487 | 0.311 | 0.246 | 0.18 |
| U-Net | 0423 | 0.307 | 0.25 | 0.209 |
| MSDM_mse | 0.437 | 0.341 | 0.29 | 0.255 |
| MSDM_ssim | **0.514** | **0.413** | **0.343** | 0.291 |

**Table 3.** Average FAR on test set with different thresholds (0.1, 1, 5, 10, 25, 40, unit: dBZ). The best score is in bold-face. The second-best score is underscored (The smaller the better).

| Model | 30 min | 60 min | 90 min | 120 min |
|---|---|---|---|---|
| Optical Flow | 0.316 | 0.391 | 0.439 | 0.474 |
| ConvLSTM | 0.265 | 0.295 | **0.242** | **0.246** |
| U-Net | 0.293 | 0.309 | 0.313 | 0.309 |
| MSDM_mse | 0.329 | 0.364 | 0.387 | 0.399 |
| MSDM_ssim | **0.237** | **0.27** | 0.303 | 0.335 |

10. Otherwise, the MSDM ranks very differently depending on the observation times (from 30 to 120 minutes) with the 0.1 dBZ threshold. Is it logical and explainable? Is it worth noticing that the MSDM ranks consistently (second best score) with the 40 dBZ thresholds. What would be the scores of the MSDM for the Radar Echo Extrapolation at other dates and times?

Response: We choose six thresholds (0.1, 1, 5, 10, 25, 40) and introduce more metrics (HSS,FAR,SSIM) to evaluate model performances. To stress the importance of areas with large radar reflectivity, we assign a weight $w($ threshold $)$ (Eq. 9) to different thresholds and calculate the weighted CSI and HSS.

Changes in the manuscript:

$$w(\text{threshold}) = \begin{cases} 1, & \text{threshold} = 0.1 \\ 1, & \text{threshold} = 1 \\ 2, & \text{threshold} = 5 \\ 3, & \text{threshold} = 10 \\ 5, & \text{threshold} = 25 \\ 8, & \text{threshold} = 40 \end{cases} \tag{9}$$

The results of scores have been shown in the previous comment.

11. L 165 -It seems that using the Modified Structural Similarity Index (denoted by SSIM) is counter-productive in terms of MAE and RMSE. Why use it? Once more,

I wonder if the example of results produced for one date and time has a general value.

Response: We train each model with MSE loss function to make comparison with the model trained with SSIM. Examples and evaluating scores have been shown in the revised manuscript. Fig 5 shows that MSE cannot predict the large-value area of radar echo.

Changes in the manuscript: We train the MSDM using MSE loss function to make comparison with the model trained with SSIM (Fig 1 and Fig 2). More explanations will be shown in the revised manuscript.

[Figure]

Figure 1 Models trained with SSIM

[Figure]

Figure 2 Models trained with MSE

12. L 186 - Figure 8 - Looking at this figure, I am not very convinced that the CSI of the Quantitative Precipitation Nowcasting are better using the random forest than using the Z-R relationship. In general, the scores are quite similar. Could the authors try to better advocate the random forest method?

Response: The CSI describe the spatial distribution of precipitation. We use RMSE and to describe the accuracy of different methods in the revised manuscript. When estimating the precipitation rate at a specific period, the number of data sample is very small, so it is not suitable for methods such as deep learning that require big data. Regressive method such as Random forest will perform better on a small sample of data. Changes in the manuscript:

[Figure]

13. L 212 - The acronyms "RNN" and "GRU" should be developed.
Response: It has been developed.
Changes in the manuscript: 'Recurrent Neural Network(RNN), Gated Recurrent Unit(GRU).'

14. My general feeling about the AI methods used separately or combined together through the paper is that all of them have advantages and drawbacks. I suggest to the authors to add a final synthetic table describing the strong points and weak points of the methods and of their combinations. This would greatly help the readers to understand the arguments of the authors.

Response: Thank you very much for this suggestion. We summarize these methods and evaluate them in terms of 12 aspects in the revised manuscript.
Changes in the manuscript: We copy the Table 4 here and the discussion and conclusion will be shown in the revised manuscripts.

**Table 4.** Evaluation on four models with (The less the better)

| | The amount of data required for training↓ | Time used for training model↓ | False Alarm Rate↓ | Accumulative system error↓ |
|---|---|---|---|---|
| Optical flow | 1 | 1 | 2 | 1 |
| ConvLSTM | 4 | 4 | 3 | 2 |
| U-Net | 2 | 2 | 4 | 2 |
| MSDM | 3 | 3 | 1 | 4 |

(The more the better)

| | The ability to capture spatial characteristics↑ | The ability to capture temporal characteristics↑ | The ability to predict initiation and decay of radar echo↑ | 0~1 hour forecast accuracy↑ |
|---|---|---|---|---|
| Optical flow | 1 | 3 | 1 | 3 |
| ConvLSTM | 2 | 4 | 2 | 1 |
| U-Net | 3 | 1 | 3 | 2 |
| MSDM | 4 | 1 | 4 | 4 |

(The more the better)

| | 1~2 hour forecast accuracy↑ | The ability to maintain the shape of radar echo↑ | Clarity of radar image↑ | Conform to the laws of physics↑ |
|---|---|---|---|---|
| Optical flow | 1 | 4 | 4 | 4 |
| ConvLSTM | 4 | 1 | 1 | 1 |
| U-Net | 3 | 2 | 2 | 2 |
| MSDM | 2 | 3 | 3 | 3 |

Referee comment 4:

1. First of all, it is unclear why the authors prefer the multi-model method when it does not give the best results in comparison with others.

Response: In the revised manuscript, we make some modifications to MSDM, and it outperforms other baseline models in most of the metrics. Our original intention to build the MSDM model is to make better use of meteorological data, as well as some existing deep learning and machine learning models. We hope to bring their advantages together through MSDM. In addition, the biggest advantage of MSDM is its transferability. Apart from satellite data, any other data (wind speed, pressure, temperature, etc.) can be used as input to the model. Meanwhile, not only the optical flow method can be used to extrapolate satellite data, but any other sequence-to-sequence model (ConvLSTM, TrajGRU, etc.) can be integrated into MSDM to extrapolate satellite data.

Changes in the manuscript: We modify the architecture of MSDM and introduce more metrics to evaluate model performances: CSI, HSS, FAR, RMSE, SSIM.

2. Furthermore, to me this method just seems like a combination of all other possible methods: Optical flow, Random forest and Convolutional Neural Network (CNN). Can one really disentangle what contributions to the solution of the problem are coming from each of the components? More insight into this is needed, especially if the method is not the one leading to the best results.

Response: Thank you very much for this suggestion. We summarize the contribution of these methods in the revised manuscript.

Changes in the manuscript: We use Table 4 to evaluate these models and discuss their advantages and drawbacks. The aim of our paper also be concluded in the 'Conclusions and dicussions'.

3. Parentheses and periods misplaced.

Response: We correct the use of parentheses and periods.

4. Incorrect double citations throughout the paper

Response: The change has been made.

Changes in the manuscript: We remove the repeated citations.

5. Many acronyms that are never defined.

Response: The acronyms have been defined

6. L11. its -> their or the

Response: The change has been made.

Changes in the manuscript: 'due to the spinup issue'

7. L12-13. check sentence

Response: We have revised the grammar of the entire paper.

8. L14. ndarray?

Response: This is a specific data format from the numpy package in python.

Changes in the manuscript: We replace 'ndarray' with 'tensor'.

9. L28. Unnecessary 'the'

Response: The change has been made.

Changes in the manuscript: Tremendous meteorological data are produced.

10. L74-75. Due to limits on computational resources

Response: The change has been made.

Changes in the manuscript: 'Due to limits on computational resources'

11. Figure 6 increase the labels It is not very scientific to label the method being tested as 'ours'

Response: The 'Ours' in figures has been replaced with 'MSDM'

Changes in the manuscript: The 'Ours' in Figure 3,5,6 have been replaced with 'MSDM'

---

## Author Response (AR2)

Comments: Black
Response to topical editor: Blue

Comments to the Author:
The reviewer comments have been addressed, but the paper does not have a good literature review. There is extensive work out there about the use of machine learning and deep learning models for precipitation and related problems. I suggest adding a dozen more references, and possibly creating a related work section that provides a comprehensive literature review. Although not required, these could be part of the literature:

Chandra, R., Cripps, S., Butterworth, N., & Muller, R. D. (2021). Precipitation reconstruction from climate-sensitive lithologies using Bayesian machine learning. Environmental Modelling & Software, 139, 105002.

Chandra, R., & Kapoor, A. (2020). Bayesian neural multi-source transfer learning. Neurocomputing, 378, 54-64.

Response: We thank the topical editor for your kind acknowledgement of our manuscript and providing part of the literature. We believe that with the help of your work, this manuscript has further improved. Thank you!

In response to the editor's comments, we add a 'Related work' section and more references to describe the use of ML and DL for radar and precipitation.